# Non-Hermitian control between absorption and transparency in perfect zero-reflection magnonics

Jie Qian[1,2], C. H. Meng[1], J. W. Rao [3], Z. J. Rao[1], Zhenghua An [1,4,5] ✉, Yongsheng Gui[2] & C. -M. Hu [2] ✉

Recent works in metamaterials and transformation optics have demonstrated exotic properties in a number of open systems, including perfect absorption/transmission, electromagnetically induced transparency, cloaking or invisibility, etc. Meanwhile, non-Hermitian physics framework has been developed to describe the properties of open systems, however, most works related to this focus on the eigenstate properties with less attention paid to the reflection characteristics in complex frequency plane, despite the usefulness of zero-reflection (ZR) for applications. Here we demonstrate that the indirectly coupled two-magnon system not only exhibits non-Hermitian eigenmode hybridization, but also ZR states in complex frequency plane. The observed perfect-ZR (PZR) state, i.e., ZR with pure real frequency, is manifested as infinitely narrow reflection dips (~67 dB) with infinite group delay discontinuity. This reflection singularity of PZR distinguishes from the resonant eigenstates but can be adjusted on or off resonance with the eigenstates. Accordingly, the absorption and transmission can be flexibly tuned from nearly full absorption (NFA) to nearly full transmission (NFT) regions.

Manipulating photon transmission/absorption in a predesigned manner while keeping zero reflection (ZR) is important for applications in general wave optics. For practical applications, the transmitted photons are critical for signal transfer and filtering, and can reveal important light-matter interaction physics[1] (such as in electromagnetically induced transparency[2]) or be encoded for information processing. The absorbed photons are crucial for signal conversion (e.g., photoelectric detection[3–7]) and energy harvesting[8] (e.g., photothermal, photoelectric, photo-spin-voltaic[9], or photomagnonic[10] conversions). The reflected photons, however, are often harmful not only because the transmission/absorption efficiency can be sacrificed, but also because reflected photons can interfere with excitation and play detrimental effects to the performance of whole optical setup (e.g., the

frequency stability of the excitation laser)[11]. Therefore, maintaining zero reflection, while simultaneously tailoring transmission/absorption is desirable for specific applications[12–14].

To this aim, tremendous[15–20] efforts have been made in several closely related fields. In metamaterials[15–17] and optomechanics[18,19], significant progresses such as perfect absorption, perfect transmission, and electromagnetically induced transparency/absorption (EIT/EIA)[18–22] have been reported. Transformation optics has demonstrated cloaking and provided us a strategy for finding ZR condition with inhomogeneous, anisotropic materials. Optical systems with parity-time symmetry attract increasing interest partly because they can suppress reflection[23]. Horsley et al. introduced Kramers-Kronig complex potentials which can approach reflectionless propagation of

[1]State Key Laboratory of Surface Physics, Institute of Nanoelectronic Devices and Quantum Computing, Department of Physics, Fudan University, Shanghai 200433, China. [2]Department of Physics and Astronomy, University of Manitoba, Winnipeg R3T 2N2, Canada. [3]School of Physical Science and Technology, Shanghaitech University, Shanghai 201210, China. [4]Shanghai Qi Zhi Institute, 41th Floor, AI Tower, No. 701 Yunjin Road, Xuhui District, Shanghai 200232, China. [5]Yiwu Research Institute of Fudan University, Chengbei Road, 322000 Yiwu City, Zhejiang, China. ✉e-mail: anzhenghua@fudan.edu.cn; Can-Ming.Hu@umanitoba.ca

photons[24]. The suppressed reflection in these works relates essentially to the rich mode interaction inside the system and their interference with incident beams. Although not explicitly mentioned, the non-Hermiticity in these works has played an important role. Meanwhile, the emergent non-Hermitian physics community has disclosed interesting mode interactions such as level repulsion (LR) and level attraction (LA). Several singular properties have also been reported, such as zero-damping states[25,26], exceptional points[27,28], skin effects[29] and so on. These singular behaviors have been observed mainly in the transmission spectra of the reported system. For example, infinitely sharp transmission dips are observed in $|S_{12}(\omega)|$ and $|S_{21}(\omega)|$ spectra in the LA region of the non-Hermitian cavity magnonics systems[25,26]. So far, however, very few works[30,31] have addressed the reflection singularity despite that ZR being critical for many applications.

Here, we propose a magnonic device with two low-damping yttrium iron garnet (YIG) spheres coupled to a same microwave transmission line (MTL). These two magnonic resonators couple only indirectly to each other being mediated by the traveling wave in MTL. By adjusting the microwave propagating delay phase ($\Phi$) between the two resonators, we can control the hybridized states covering both level attraction/repulsion (LA/LR). We explicitly show that the ZR condition exists generally in the complex frequency plane and distinguishes from the resonant eigenstates of the effective non-Hermitian Hamiltonian. When ZR frequency becomes purely real, perfect ZR (PZR) becomes observable in our experiments as sharp reflection dips. This reflection singularity of PZR is attributed to the destructive interference between the direct and high-order reflected waves. It can be regulated to coincide with or deviate from the eigenmodes and accordingly the absorption/transmission of the coupled system can be managed. Our work expands the non-Hermitian physics in magnon-photon coupled systems and sheds light to the practical applications such as microwave circuits, photonic chips, and quantum information.

## Results and discussion
### Structure of the indirectly coupled two-magnon system
In our device (Fig. 1a) two 1-mm diameter yttrium iron garnet spheres (YIG 1 and YIG 2) are placed above a transmission line (MTL) with a distance of $l = 25$ mm. Microwave signals are loaded from either port 1 or port 2 to excite ferromagnetic resonance (FMR) in the two YIG spheres. An external magnetic field (**H**-field) is applied perpendicularly to the waveguide plane, which saturates the magnetizations of the two YIG spheres and determines the resonant frequencies of two magnon modes, $\omega_1$ and $\omega_2$. To fine tune their frequency detuning $\Delta_H = \omega_1 - \omega_2$, an additional small coil is placed under the position of YIG 1 so that its local magnetic field ($\delta H \ll H$) can be precisely adjusted. As a result, two nonidentical resonators with excellent tunability of resonant frequencies ($\omega_1$ is determined by $\mathbf{H} + \delta \mathbf{H}$ and $\omega_2$, by $\mathbf{H}$) are configured. The very high accuracy (~0.1 MHz) in fine parameter tuning of $\Delta_H$ enables the study of singular behaviors in spectrum of the non-Hermitian coupled systems[32,33].

As shown in Fig. 1b, the excited resonators dissipate their energy through intrinsic damping rates ($\gamma_{1,2}$), which are typically low and depend on the natural properties of YIG spheres including the surface roughness, impurities and defects, etc. Besides that, two resonators couple to the same dissipative channel (MTL)[34,35] with the extrinsic damping rates of $\kappa_{1,2}$, their cooperative effect would induce an effective indirect coupling $-i\Gamma e^{i\Phi}$, where $\Gamma = \sqrt{\kappa_1 \kappa_2}$ and $\Phi = 2\pi l/\lambda$. In contrast to earlier works, since $l$ in our system is large enough to suppress the direct coupling between two resonators, the indirect coupling mediated by propagating photons dominates the system properties. Depending on the value of $\Phi$, the indirect coupling strength can be either purely real (coherent coupling), purely imaginary (dissipative coupling), or mixed real and imaginary contributions. Experimentally, $\Phi$ can be adjusted by either $\lambda$ or $l$. Here, we choose the non-mechanical method[33] and change wavelength $\lambda$ at a fixed distance $l$ to avoid mechanical change of YIG spheres which may induce unwanted influence to the external damping rates $\kappa_{1,2}$. When the **H**-field is swept over a large range, the resonant frequencies, and thereby, $\lambda$ are modulated accordingly.

### Hybridized eigenstates in the two-magnon non-Hermitian system
Under the rotating-wave approximation, the effective non-Hermitian Hamiltonian of the subsystem can be created by using Feshbach

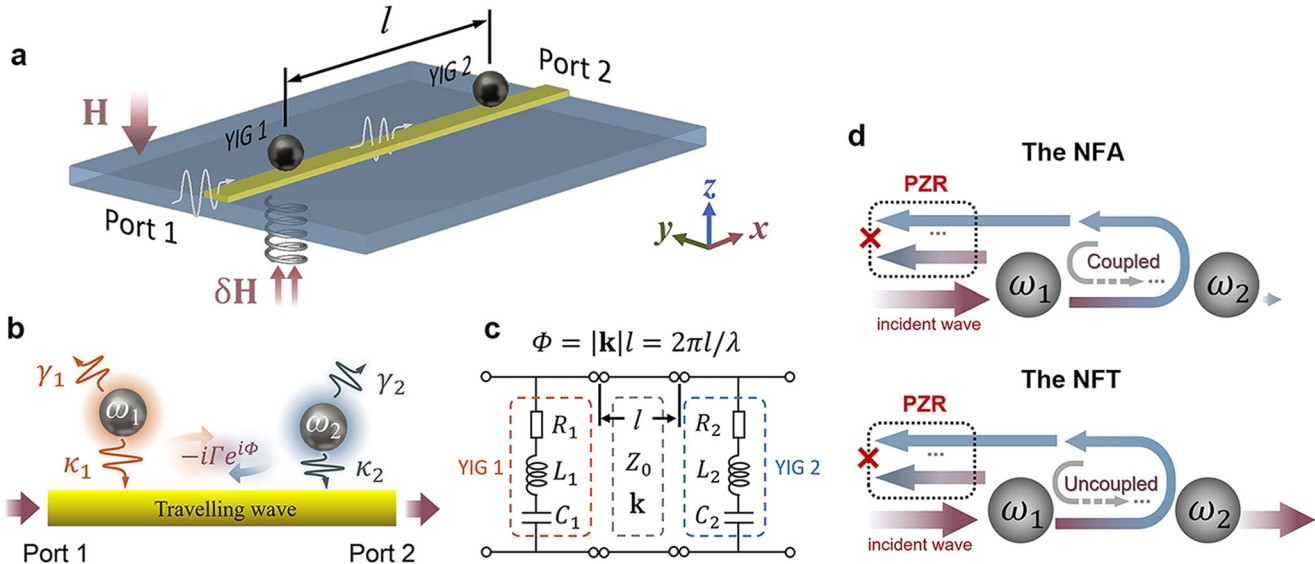

**Fig. 1 | Indirectly coupled two-magnon system. a** Schematic diagram of the experimental setup. Two separated YIG spheres ($l = 25$ mm) coupled to a transmission line are placed under a vertical magnetic field (**H**-field), the local magnetic field (**H** + $\delta$**H**) around the YIG 1 is controllable independently by finely adjusting coil current. **b** Schematic diagram showing the two resonators are side-coupled to a transmission line and indirectly interact with each other mediated by traveling wave with the effective coupling strength of $-i\Gamma e^{i\Phi}$. **c** Equivalent circuit of the coupled system. Circuit elements used to model the two YIG spheres are marked by the orange and blue box, respectively. **d** Schematic diagram showing the nearly full absorption (NFA) and nearly full transmission (NFT) under the perfect zero-reflection (PZR) condition.

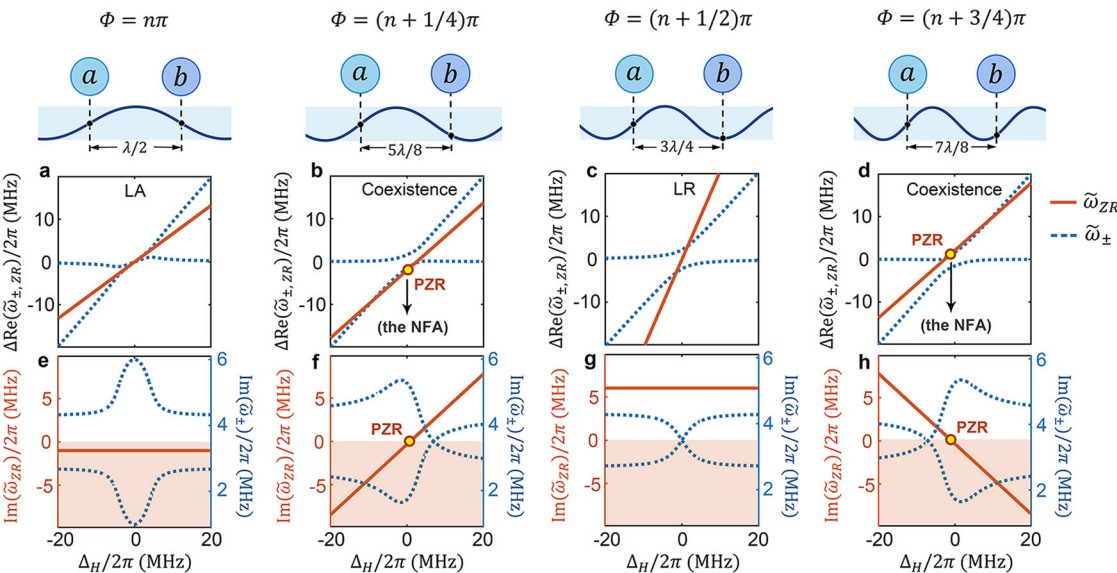

**Fig. 2 | The theoretical calculations of zero-reflection (ZR) condition and mode hybridization. a–d** The frequency detunings of the real parts of ZR conditions, $\Delta\mathrm{Re}(\tilde{\omega}_{ZR}) = \mathrm{Re}(\tilde{\omega}_{ZR}) - \omega_2$ (orange solid lines), and the frequency detunings of the real parts of eigenvalues, $\Delta\mathrm{Re}(\tilde{\omega}_\pm) = \mathrm{Re}(\tilde{\omega}_\pm) - \omega_2$ (blue dashed curves), as functions of $\Delta_H$ in four $\Phi$ cases, $\Phi = n\pi$, $(n+1/4)\pi$, $(n+1/2)\pi$, $(n+3/4)\pi$, $n \in N$, respectively. **e–h** Correspondingly, the imaginary parts of ZR conditions, $\mathrm{Im}(\tilde{\omega}_{ZR})$ (orange solid lines) and the frequency detunings of the imaginary parts of eigenvalues,

$\mathrm{Im}(\tilde{\omega}_\pm)$ (blue dashed curves), as functions of $\Delta_H$. In (**e–h**), $\mathrm{Im}(\tilde{\omega}_{ZR})$ correspond to the left scale and $\mathrm{Im}(\tilde{\omega}_\pm)$ correspond to the right scale, the white and orange areas are divided by the horizontal line of $\mathrm{Im}(\tilde{\omega}_{ZR}) = 0$. Circles in (**b, f**) and (**d, h**) represent the perfect zero-reflection (PZR) conditions. The top panels of (**a–d**) are the schematic diagrams at $\Delta_H = 0$ when $\Phi = \pi$, $5\pi/4$, $3\pi/2$, $7\pi/4$, respectively, assuming $n = 1$. The abbreviations LA and LR represent level attraction and level repulsion, respectively.

projection approach[36] (see Supplementary Note 1 for details),

$$\mathcal{H}_{eff} = \hbar\tilde{\omega}_1\hat{a}^\dagger\hat{a} + \hbar\tilde{\omega}_2\hat{b}^\dagger\hat{b} - \hbar(i\Gamma e^{i\Phi})(\hat{a}^\dagger\hat{b} + \hat{b}^\dagger\hat{a}), \quad (1)$$

where $\hat{a}^\dagger$ ($\hat{a}$) and $\hat{b}^\dagger$ ($\hat{b}$) represent the creation (annihilation) operators of the two magnon modes, obeying the commute relations of $[\hat{a},\hat{a}^\dagger] = 1$, $[\hat{b},\hat{b}^\dagger] = 1$, and $[\hat{a},\hat{b}] = 0$. $\tilde{\omega}_{1,2} = \omega_{1,2} - i(\gamma_{1,2} + \kappa_{1,2})$ are the complex frequencies of the uncoupled magnon modes, with the imaginary parts being related to the experimentally observed frequency linewidths. Similar non-Hermitian Hamiltonian has been used in various area including atomic system[37], cavity quantum electrodynamics[34], optomechanical systems[35], and so on. The eigenvalues of Eq. (1) can be solved to be

$$\tilde{\omega}_\pm = \frac{1}{2}\left[\tilde{\omega}_1 + \tilde{\omega}_2 \pm \sqrt{(\tilde{\omega}_1 - \tilde{\omega}_2)^2 + 4(i\Gamma e^{i\Phi})^2}\right], \quad (2)$$

where the real parts of $\tilde{\omega}_\pm$ give the dispersions of two hybridized modes, and the imaginary parts $\mathrm{Im}(\tilde{\omega}_\pm)$ correspond to their damping rates. The mode hybridization appears to be similar to previous works[38–40], covering both level attraction (LA) and level repulsion (LR) regions as shown in Fig. 2a–d with four representative cases, $\Phi = n\pi$, $(n+1/4)\pi$, $(n+1/2)\pi$, $(n+3/4)\pi$ ($n \in N$), respectively. On the other hand, our work distinguishes from earlier works in the two aspects:

First, traveling photon bath in this work always mediates the indirect coupling between two resonators and leads to both LA and LR; while in most of previous works, LR is often attributed to the direct coupling between resonators with large spatial wavefunction overlapping. The interaction mediated by a same photon bath here allows to reveal the essential difference between LA and LR: When $\Phi = n\pi$, two magnon modes are synchronized (first cartoon in Fig. 2) by traveling photons with either in-phase ($n$ is even) or 180-degree out-of-phase ($n$ is odd) spin precessions in the two YIG spheres. This results in the distinct feature of a dissipatively coupled system that the hybridized modes attract each other (blue dashed curves in Fig. 2a) while the

corresponding imaginary parts repulse (blue dashed curves in Fig. 2e). In contrast, when $\Phi = (n+1/2)\pi$, two magnon modes have a phase difference of $\pi/2$ (third cartoon in Fig. 2), the hybridized modes show the distinct feature of a coherently coupled system where the real parts repulse each other (Fig. 2c) while the corresponding imaginary parts attract (Fig. 2g), in this situation, the coupled system behaves as if magnons keep hopping in-between two YIG spheres, although this long-range hopping of magnons cannot occur straightforwardly like conventional Rabi oscillation, but has to be mediated by traveling photons in MTL. For other $\Phi$ values, the synchronization and hopping coexist in the system thus lead to the mixed interaction of LA and LR, as shown in Fig. 2b, f and d, h, for $\Phi = (n+1/4)\pi$ and $(n+3/4)\pi$, respectively. Interestingly, the complex eigenvalues in Eq. (2) satisfy the unit round-way propagation of the trapped photons in-between two YIG spheres, implying the Fabry-Pérot (FP)-like resonant mechanism in our device (see Supplementary Note 1). In contrast to conventional FP cavity with simple dispersionless mirrors, the accumulated phase $\Phi_{\mathrm{round}} = 2n\pi$ has to take into account the reflection phase shift by both YIG 1(2), $\phi_{1(2)}$ in addition to the propagating phase $\Phi$. Namely, $\Phi_{\mathrm{round}} \equiv \phi_1 + \phi_2 + 2\Phi = 2n\pi$ and this formula applies for single round-trip propagation of microwave photons but is found to be sufficient for reproducing our following experimental data (see Supplementary Note 1). Compared to previously reported Fabry-Pérot (FP) type interaction with identical resonant "mirrors" ($\omega_1 = \omega_2$, $\Phi \neq 0$)[41,42], our system represents a more generalized coupling model with nonidentical resonant mirrors and arbitrary $\Phi$ which therefore unifies the FP-type and Friedrich-Wintgen (FW) type ($\omega_1 \neq \omega_2$, $\Phi = 0$) interactions[43].

Second, different from previous works using two different types of resonators, namely, metal cavity resonator and magnonic resonator[25,26,38], the two same magnonic-type resonators in this work allows them more balanced in terms of the intrinsic and extrinsic damping rates ($\gamma_1, \gamma_2, \kappa_1, \kappa_2$). The balanced dampings, in particular $\kappa_1 \sim \kappa_2$, are important for establishing the balanced FP-like resonance since the reflection at the YIG 1(2) is proportional to the corresponding $\kappa_{1(2)}$ (see Supplementary Note 1). This balanced FP-like resonance allows the ZR

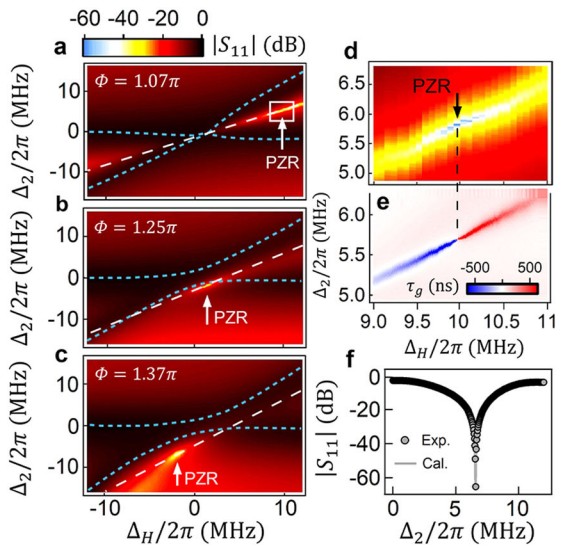

**Fig. 3 | Perfect zero-reflection (PZR) observed in experiment. a–c** Mappings of |$S_{11}$| as functions of $\Delta_H$ and $\Delta_2$ in four cases: fixing $\omega_2/2\pi = 3.6$ GHz, 4.24 GHz, 4.69 GHz, respectively. **d** High-resolution zoom-in image of (**a**). **g–i** Mappings of |$S_{22}$| as functions of $\Delta_H$ and $\Delta_2$ in four $\omega_2$ cases. Blues dotted curves are the fitted hybridized modes, white long dashed lines are the calculated frequencies of the ZR states. The PZR conditions are marked by arrows. **j** High-resolution zoom-in image of (**g**). **e** and **k** are the group delay $\tau_g$ mappings corresponding to (**d**) and (**j**), the PZRs emerge as singularities, where the group delay abruptly switches between negative infinity and positive infinity. **f** and **l** are the PZR spectra (|$S_{11}$| and |$S_{22}$|) at the black arrows marked fields in (**d**) and (**j**), respectively.

condition can be observed near the coupling region, as will be seen in the next sections.

## Zero-reflection (ZR) states in complex frequency plane of the non-Hermitian system

The reflection ($S_{11}$ and $S_{22}$) and transmission ($S_{21}$ and $S_{12}$) spectra can be derived from the effective Hamiltonian of Eq. (1) based on coupled-mode theory, or alternatively from the equivalent circuit model (Fig. 1c) using ABCD matrix[44] method (see Supplementary Note 1 for details). These four S-parameters can be derived as

$$S_{11}(\omega) = -2i\omega \frac{\kappa_1(X_2 + 2i\omega\kappa_2) + \kappa_2 e^{2i\Phi}(X_1 - 2i\omega\kappa_1)}{(X_1 + 2i\omega\kappa_1)(X_2 + 2i\omega\kappa_2) + 4\omega^2\kappa_1\kappa_2 e^{2i\Phi}}, \quad (3a)$$

$$S_{22}(\omega) = -2i\omega \frac{\kappa_1 e^{2i\Phi}(X_2 - 2i\omega\kappa_2) + \kappa_2(X_1 + 2i\omega\kappa_1)}{(X_1 + 2i\omega\kappa_1)(X_2 + 2i\omega\kappa_2) + 4\omega^2\kappa_1\kappa_2 e^{2i\Phi}}, \quad (3b)$$

$$S_{21}(\omega) = S_{12}(\omega) = \frac{X_1 X_2 e^{i\Phi}}{(X_1 + 2i\omega\kappa_1)(X_2 + 2i\omega\kappa_2) + 4\omega^2\kappa_1\kappa_2 e^{2i\Phi}}, \quad (4)$$

where $X_{1,2} = \omega^2 - \omega_{1,2}^2 + 2i\omega\gamma_{1,2}$. Under the rotating-wave approximation, $\omega^2 - \omega_1^2 \sim 2\omega(\omega - \omega_1)$ and $\omega^2 - \omega_2^2 \sim 2\omega(\omega - \omega_2)$ can be obtained. Note that, in the complex frequency plane, poles[28] and zeros of the reflection ($S_{11}$ and $S_{22}$ in Eq. (3)) correspond to the resonant eigenvalues ($\tilde{\omega}_\pm$ in Eq. (2)) and the ZR states, respectively. The balanced damping rates together with the fine parameter tuning of $\Delta_H = \omega_1 - \omega_2$ allow us to observe both the eigenmodes (peaks in $S_{11}$ and $S_{22}$) and the ZR modes (dips in $S_{11}$ and $S_{22}$) in the same parameter space and range. In Fig. 2, the solutions of ZR, i.e., $S_{11}(\omega) = 0$, are plotted by orange solid lines in complex frequency plane (Supplementary Note 2), the real parts of $\mathrm{Re}(\tilde{\omega}_{ZR})$ as functions of $\Delta_H$ are shown in Fig. 2a–d, and the corresponding imaginary parts $\mathrm{Im}(\tilde{\omega}_{ZR})$ are shown in Fig. 2e–h. $\mathrm{Im}(\tilde{\omega}_{ZR})$ approaches two constant values but with opposite signs (one positive and one negative) at two conditions: in the case of pure LA with $\Phi = n\pi$, $\mathrm{Im}(\tilde{\omega}_{ZR}) = -i\frac{\gamma_1\kappa_2 + \gamma_2\kappa_1}{\kappa_1 + \kappa_2} = -i$ MHz (Fig. 2e, left scale) and in the case of pure LR with $\Phi = (n+1/2)\pi$, $\mathrm{Im}(\tilde{\omega}_{ZR}) = i\frac{\gamma_1\kappa_2 - \gamma_2\kappa_1 - 2\kappa_1\kappa_2}{\kappa_1 - \kappa_2} = 6i$ MHz (Fig. 2g, left scale). For other $\Phi$ values, the lines of $\mathrm{Im}(\tilde{\omega}_{ZR})$ are tilted (Fig. 2f, h), meanwhile, in Fig. 2b, d, the lines

of $\mathrm{Re}(\tilde{\omega}_{ZR})$ intersect with $\mathrm{Re}(\tilde{\omega}_\pm)$, which imply the correlation between the eigenmodes and ZR modes can be tuned by $\Phi$ and $\Delta_H$.

Experimentally, we fix $\omega_2/2\pi = 3.6$ GHz, 4.24 GHz, 4.69 GHz, respectively, and sweep $\omega_1$ to obtain the reflection mappings as shown in Fig. 3. Reflection data at three different cases of $\omega_2$ are acquired and compared directly with the theoretical calculations: $\Phi = 1.07\pi$, $1.25\pi$, $1.37\pi$, respectively. Here, the values of $\Phi$ are obtained by fitting the dispersion of reflection mappings in Fig. 3 according to Eq. (2) (also see Supplementary Note 3, fitted curves represented by dashed lines in FIG. S3). The correlation between the real parts of eigenmodes ($\tilde{\omega}_\pm$) and ZR modes ($\tilde{\omega}_{ZR}$) can be confirmed in the reflection spectra (both |$S_{11}$| and |$S_{22}$|), the experimental reflection peaks agree well with the calculated eigenvalues (blue dotted curves) for all the spectra (|$S_{11}$| in Fig. 3a–c and |$S_{22}$| in Fig. 3g–i), and the reflection dips match the calculated ZR lines ($\mathrm{Re}(\tilde{\omega}_{ZR})$ plotted by the white long dash lines). As a result, the final reflection spectra are always superimposed with both eigenmodes (peaks) and ZR modes (dips).

## Singularity of perfect zero-reflection (PZR)

For $\Phi \neq n\pi/2$, the tilted ZR line (orange line in Fig. 2f or h) intersects with the real frequency axis and a critical value of $\mathrm{Im}(\tilde{\omega}_{ZR}) = 0$ can be obtained (marked by circle in Fig. 2). This critical condition is termed as perfect zero-reflection (PZR) in order to be distinguished from the general ZR state with nonzero imaginary part, $\mathrm{Im}(\tilde{\omega}_{ZR}) \neq 0$. In Figs. 2e and 2g, the horizontal orange lines show $\mathrm{Im}(\tilde{\omega}_{ZR})$ become nonzero constants in pure LA (Fig. 2e) and pure LR (Fig. 2g) regions, $\mathrm{Im}(\tilde{\omega}_{ZR}) \neq 0$ indicates there is no PZR exist. We notice that $\mathrm{Im}(\tilde{\omega}_{ZR}) < 0$ in pure LA region and $\mathrm{Im}(\tilde{\omega}_{ZR}) > 0$ in pure LR region, while in the general hybrid region (Figs. 2f and 2h) with $\Phi \neq n\pi/2$, there exists the critical point where the sign of $\mathrm{Im}(\tilde{\omega}_{ZR})$ changes abruptly leading to $\mathrm{Im}(\tilde{\omega}_{ZR}) = 0$, ensuring the general existence of PZR. The PZR condition can be derived analytically to be

$$\omega_{PZR}^\pm = \frac{1}{2} \frac{\sin(2\Phi)(\omega_1^2 - \omega_2^2)}{(\kappa_2 - \kappa_1)(\cos 2\Phi - 1) \pm \left[(\gamma_1 + \gamma_2)\cos 2\Phi + \frac{\kappa_1\gamma_2}{\kappa_2} + \frac{\kappa_2\gamma_1}{\kappa_1}\right]}, \quad (5)$$

the superscript "±" represents the case that microwave signals are loaded from port 1 (|$S_{11}(\omega)$| = 0) and from port 2 (|$S_{22}(\omega)$| = 0),

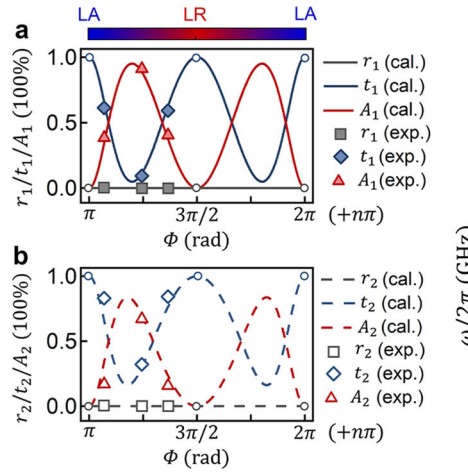
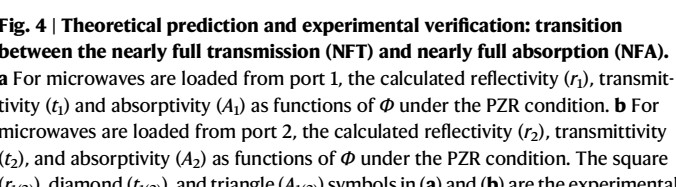
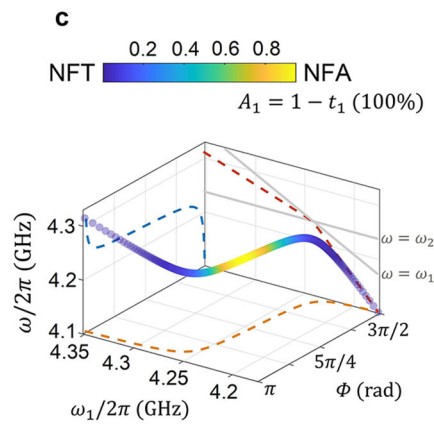

**Fig. 4 | Theoretical prediction and experimental verification: transition between the nearly full transmission (NFT) and nearly full absorption (NFA).** **a** For microwaves are loaded from port 1, the calculated reflectivity ($r_1$), transmittivity ($t_1$) and absorptivity ($A_1$) as functions of $\Phi$ under the PZR condition. **b** For microwaves are loaded from port 2, the calculated reflectivity ($r_2$), transmittivity ($t_2$), and absorptivity ($A_2$) as functions of $\Phi$ under the PZR condition. The square ($r_{1(2)}$), diamond ($t_{1(2)}$), and triangle ($A_{1(2)}$) symbols in (**a**) and (**b**) are the experimental results. Color bar shows the hybridized states for different values of $\Phi$, the abbreviations LA and LR represent level attraction and level repulsion, respectively. Open circles plotted in (**a**, **b**) indicate the PZRs do not exist at these points. **c** The calculated three-dimensional plot of the absorption/transmission as a function of $\Phi$, $\omega_1$, and $\omega$. Here, $\gamma_1/2\pi = \gamma_2/2\pi = 1$ MHz, $\kappa_1/2\pi = 1.7$ MHz, $\kappa_2/2\pi = 3.3$ MHz are set based on the fitting results of experimental spectra.

respectively. It's worth noting that, in Eq. (5), $\omega_1$ is uniquely determined by $\omega_2$ due to the requirement of $\mathrm{Im}(\tilde{\omega}_{ZR}) = 0$ (see Supplementary Note 2), thus the PZR appears at a certain frequency point when the external magnetic field fixed. The underlying physical picture of the PZR condition is the destructive interference (Fig. 1d) between direct reflection by the first YIG sphere and the multiple reflection involving the second YIG sphere leads to the vanishing reflection amplitude (Supplementary Note 1). Since the strictly destructive interference occurs only at a single frequency, the linewidth of the PZR spectra becomes infinitely small with an effectively ultra-high Q-factor, for instance, ultra-sharp dip about 67 dB is observed with the measuring accuracy of ~0.1 MHz (Fig. 3f or Fig. 3l). Earlier work by Mukhopadhyay et al.[30] proposed similar physics behind PZR by coupling a number of identical and lossless resonators to a common waveguide. Here, Eq. (5) provides a more general formalism for nonidentical and lossy resonators with traveling-photon-mediated indirect coupling.

Experimentally the general existence of the PZR condition has been verified. As can be seen from Fig. 3, yellow sharp dips appear in the reflection spectra. Figure 3a–c are the experimental mappings of $|S_{11}|$ as a function of $\Delta_H = \omega_1 - \omega_2$ and $\Delta_2 = \omega - \omega_2$ for the cases of $\Phi = 1.07\pi$, $1.25\pi$, $1.37\pi$, respectively, in which sharp reflection dips of the PZRs are clearly visible at the fields marked by arrows, $\Delta_H^+/2\pi = 9.9, 0.9, -2.2$ MHz. These values agree consistently with the theoretical prediction by Eq. (5). Similarly, in the experimental mappings of $|S_{22}|$ (Fig. 3g–i), sharp reflection dips show up again despite that the quantitative $\Delta_H^-$ values of the PZRs are different due to the asymmetric parameters of the two YIG spheres, i.e., $\kappa_1 \neq \kappa_2$. In Fig. 3g–i, the PZR occurs at the field $\Delta_H^-/2\pi = -11.0, -4.1, -4.8$ MHz, respectively, which is also in good accord with Eq. (5). Figure 3d (3j) is the enlarged view of the PZR region in Fig. 3a (3g). Figure 3e (3k) is the group delay $\tau_g$ mapping corresponding to Fig. 3d (3j), $\tau_g = -\partial P_{11(22)}/\partial\omega$ is defined as the negative derivative of the reflection phase $P_{11(22)}$. Beside the ultra-sharp dip (~67 dB with ~0.1 MHz measuring accuracy) observed in the reflection spectrum (Fig. 3f (3l)), an additional intriguing feature of the PZR is the group delay $\tau_g$, which is defined as the negative derivative of the reflection phase $P_{11(22)}$, $\tau_g = -\partial P_{11(22)}/\partial\omega$. As revealed by Fig. 3e (3k), the PZR emerges in the position where $\tau_g$ switches abruptly between negative infinity and positive infinity, $\tau_g \to \pm\infty$, so that the group velocity $v_g = 1/\tau_g$ becomes zero. This suggests

that the PZR observed in our system is indeed a singularity and may be utilized for slowing microwave photons[27].

## Transmission and absorption under the PZR condition

For arbitrary value of $\Phi$, once the PZR condition of Eq. (5) is satisfied, the nearly full absorption (NFA) and nearly full transmission (NFT) can be realized by trading-off the absorption and transmission ($A_{1(2)} + |S_{21(12)}|^2 = 1$) and experimentally, by cooperatively adjusting $\Phi$ and $\Delta_H$. On the premise of satisfying Eq. (5), the Eq. (4) can be written as

$$S_{11}(\omega_{PZR}^+) = 0, \quad S_{21}(\omega_{PZR}^+) = \frac{e^{i\Phi}}{1 + \frac{2i\omega_{PZR}^+ \kappa_2 (1 - e^{2i\Phi})}{(\omega_{PZR}^+)^2 - \omega_2^2 + 2i\omega\gamma_2}}, \quad (6a)$$

$$S_{22}(\omega_{PZR}^-) = 0, \quad S_{12}(\omega_{PZR}^-) = \frac{e^{i\Phi}}{1 + \frac{2i\omega_{PZR}^- \kappa_1 (1 - e^{2i\Phi})}{(\omega_{PZR}^-)^2 - \omega_1^2 + 2i\omega\gamma_1}}. \quad (6b)$$

We have numerically analyzed the transmission and absorption by Eq. (6), as shown in Fig. 4a, b, the transmittivity $t_{1(2)} = |S_{21(12)}|^2$ (blue curves) and absorptivity $A_{1(2)} = 1 - |S_{21(12)}|^2$ (red curves) show complementary oscillation tendency with $\Phi$. (Gray lines represents $r_{1(2)} = |S_{11(22)}|^2 = 0$ due to the PZR). The color bar shows the hybridized states at different $\Phi$ values, at the purely coherent (LR) and purely dissipative (LA) coupling regions ($\Phi = n\pi/2$, $n \in N$), the PZR does not exist, and hence the full transmission cannot be reached strictly but can be approached infinitely when $\Phi \to n\pi/2$. On the other hand, in the region of LR and LA coexistence, the nearly full absorption (NFA) can be approached when $\Phi$ gets close to (but still different from) $n\pi/2 + \pi/4$. All experimental datapoints depicted by symbols in Fig. 4a, b show excellent agreement with the numerical results (detail measurement data see Supplementary Note 3, Fig. S5). Since in the two-port configuration[45], the PZR ($S_{11(22)} = 0$) and zero-transmission ($S_{21(12)} = 0$) cannot be satisfied simultaneously and therefore the full absorption cannot be realized strictly. In Fig. 4a, the achievable maximum absorption under the PZR is ~96% when microwave signals are loaded from port 1, and in Fig. 4b, when microwave signals are loaded from port 2, the achievable maximum absorption is ~80%. This implies to realize the NFA, $\kappa_2 > \kappa_1$ is necessary if microwave signals are loaded from port 1 and vice versa. By increasing the ratio of $\kappa_2/\kappa_1$

(see Supplementary Note 4 for detail), the maximum absorption of $A_1$ can be approached to unity (while $A_2$ is sacrificed). In the limit of $\kappa_2/\kappa_1 \to \infty$, the two-port configuration evolves into one-port configuration where the full absorption condition exists. Figure 4c displays a three-dimensional plot of the absorption/transmission when sweeping $\omega_1$ and $\Phi$, while keeping $\omega_2/2\pi = 4.24$ GHz. The nearly full absorption appears in the small detuning region ($\omega_1 \cong \omega_2$) as colored by yellow, and the nearly full transmission takes place in the large detuning region. As shown schematically in Fig. 1d, in the nearly full absorption (NFA) region ($\omega_1 \cong \omega_2$), FP-like mechanism for photons trapped between the two resonators assists the magnon exchange between two YIG spheres, and eventually, most of the microwave energy is dissipated through the spin processions in the two YIG spheres despite the intrinsic losses of two magnon modes are rather small (~1 MHz). While in the nearly full transmission (NFT) region, the large frequency detuning ($\omega_1 \neq \omega_2$) makes two magnon modes are nearly uncoupled with each other at the PZR frequency, so that little microwave energy can be absorbed by YIG spheres since $\omega_{PZR} \neq \omega_1$ and $\omega_{PZR} \neq \omega_2$. From Fig. 4c, in the negative detuning region ($\omega_1 < \omega_2$), $\omega_{PZR}$ increases as $\omega_1$ increases with an approximate slope of $\kappa_2/(\kappa_2 - \kappa_1)$. In the positive detuning region ($\omega_1 > \omega_2$), the slope changes to $\kappa_2/(\kappa_1 + \kappa_2)$, this result provide us a guidance to manipulate the microwave propagating and absorption for any target frequency. To sum up, by adjusting **H**-field (affects propagating delay phase $\Phi$) and $\delta$**H**-field through coil current (affects frequency detuning $\Delta_H = \omega_1 - \omega_2$), we demonstrate the microwave behavior at the PZR frequency is controllable and realize the waveguide magnonic device allowing the continuous control between nearly full transmission (infinitely close to 100%) and nearly full absorption (~96%).

In conclusion, we report the non-Hermitian control of reflectionless photon propagation, from nearly full transmission to nearly full absorption regions in a magnon-photon coupled system. By freely adjusting the frequency detuning between the two magnonic resonators and the microwave propagating delay phase, the PZR condition can be always achieved with tailored transmission/absorption. The flexible tunability and the general existence of the reflection singularity in our non-Hermitian magnonic device make the system promising for actual applications, such as lossless signal conversion and avoiding signal disturbances. Our work reveals the rich non-Hermitian physics in magnon-photon coupled systems, and meanwhile suggests the broad prospect of non-Hermitian control in the fields of microwave circuits, photonic chips, wave optics, and future quantum information technologies.

## Methods

### Device description
The waveguide magnonic device consists of a transmission line and two YIG spheres. The transmission line with a width of 1.14 mm and a length of 50 mm is fabricated on a 0.762-mm-thick Rogers RO4350B substrate. The dimensions of the substrate are $20 \times 50$ mm with a copper thickness of 35 μm on both sides. In the measured frequency range (3–5 GHz), the insertion loss of the bare waveguide is less than 0.5 dB, and the reflection keeps below −20 dB. Two 1 mm diameter YIG spheres separated by 2.5 mm are glued to the end of a displacement cantilever and adjusted accurately close to the copper waveguide through a three-dimensional adjustable stage. The crystallographic directions of two YIG spheres are not identified but remain fixed in all experiments. The magnon mode in the YIG sphere follows the Kittel dispersion equation $\omega_m = \gamma_e \mu_0 (|\mathbf{H}| + |\mathbf{H}_0|)$, where $\mu_0$ is the vacuum permeability. For the YIG 1, the gyromagnetic ratio and the anisotropy field are $\gamma_e/2\pi = 30$ GHz/T and $\mu_0|\mathbf{H}_0| = 7.9$ mT, respectively. For the YIG 2, $\gamma_e/2\pi = 29$ GHz/T and $\mu_0|\mathbf{H}_0| = 8.4$ mT. The static magnetic field is applied perpendicularly to the waveguide plane.

### Measurement setup
The reflection and transmission spectra are measured using a vector network analyzer (VNA) with an input power of −5 dBm.

## Data availability
The data that support the findings of this study are available within the article and its Supplementary Information. Additional relevant data are available from the corresponding authors upon reasonable request.

## Code availability
The computer codes that support the plots within this paper and the findings of this study are available from the corresponding author upon reasonable request.

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

## Acknowledgements

This work was funded by NSERC Discovery Grants and NSERC Discovery Accelerator Supplements (C.-M.H.). Z.A. acknowledges the financial support from the National Natural Science Foundation of China under Grant Nos. 12027805/11634012/11991060/11674070, the National Key Research Program of China under Grant No. 2016YFA0302000, and the Shanghai Science and Technology Committee under Grant Nos. 18JC1420402, 18JC1410300, 20JC1414700, and 20DZ1100604. J. Qian was supported in part by the China Scholarship Council (CSC). We thank Yipu Wang for the discussion and providing conceptual advice.

## Author contributions

Y.S.G. conceived the idea and designed the experiments. J.W.R. prepared the samples and built up the experiment setup. Q.J. performed all the experiments. C.H.M. contributed to the derivation of effective Hamilton with Feshbach projection approach. Q.J., Y.S.G., Z.A., and C.-M.H. performed the data analysis and co-wrote the manuscript with comments from all authors. Z.J.R. provides help in data analysis and discussion. C.-M.H. and Z.A. supervised the project.

## Competing interests

The authors declare no competing interests.
