## [Peer Review File · Nature Communications]

Reviewers' Comments:

Reviewer #1:

Remarks to the Author:

The authors have presented a very interesting work on cavity magnonics where two YIG spheres are coupled to a same microwave transmission line resonator. Several things have been shown here including indirect coupling between the two samples, and level repulsion and level attraction. While the results are interesting, I do not feel this paper meets the criteria for Nature Communications:

1. The results are interesting but there is a lot of superfluous language and terms that do not add to the discussion, instead they take away from the main findings. For instance, from the title with 'Transparency in Reflectionless' magnonics. I am unsure what this means. Are the authors quantifying transparency and reflection of microwaves (or the traveling waves) in their devices? Does not seem to be the case as they are working with the hybridisation. If they want to quantify the traveling wave behaviour, shouldn't an electromagnetic theory with the full calculation of these be needed?
2. The authors' results seem very similar to their own previous work. There is a vast literature now on dissipative coupling, which some of the authors are involved in, and this seems to be an extension of that.
3. Related to 2, the authors do not contextualise their data in terms of previous literature. In fact, the work is presented as almost unrelated to previous mechanisms which does not seem to be the case.
4. Related to 3. In Figure 3, the results seem virtually identical to the 'level-merging' behavior discussed by Boventre et al <https://doi.org/10.1103/PhysRevResearch.2.013154>. What is the relationship between a two port cavity system and this one. Seems to be deeply related but again, relationship to previous literature is missing.
5. Also related to 2. How does level attraction relate to the Lenz effect as discussed by Harder et al <https://doi.org/10.1103/PhysRevLett.121.137203>?

Finally, it is my feeling that the authors would need to address points above before this paper is considered for publication.

Reviewer #2:

Remarks to the Author:

The results presented here are very interesting, and show evidence of tunable control of a system set up with coherent magnonic resonators (based on YIG spheres) to yield interfering systems whose reflection, transmission and absorption features are tunable. The authors tune the system to one of very low reflectivity and then show that it is possible to control whether the system is transmissive or absorbing by changing the phase between the two oscillators (using a tuning field).

The paper itself is more difficult to read than it should be, due to the extensive use of specific jargon, the presentation of some statements that are confusing, a lack of clarity about when experimental and when theoretical results are being discussed, and missing physical pictures to help the reader understand intuitively the features. If these are addressed this paper is likely a significant contribution to the fundamental exploration of microwave transmission, absorption, and reflection control, and could be appropriate for Nature Communications.

1) PA, PT, and PZR are unfortunately confusing, since PZR means perfect zero reflection (hence reflection = 0) whereas PA and PT mean perfect absorption and perfect transmission (i.e. =1 not =0). The terms are also unfortunate since nothing is ever "perfect". Even $A+T+R$ is not precisely equal to 1, since there is potentially scattering that would occur out of the waveguide.

2) a statement like "perfect zero reflection is a prerequisite" is troubling since again there is always going to be some imperfect tuning and some reflection. The authors should provide bounds for these features rather than saying "negligible" or "zero" so the readers know to what level of accuracy these statements are experimentally valid.

3) it is often difficult to tell what pieces are theoretical and what pieces are experimental. Many figures are presented without this clarity (i.e. fig. 2) and only later is it apparent that this is entirely theoretical. The authors should be upfront about which results are theoretical and which experimental.

4) a key insight that would help the reader is - for perfect absorption, where is the energy going? the authors have emphasized the negligible loss of the YIG spheres, but is that loss what is driving the absorption? I wonder if the effect of the structure is to maintain a huge microwave field in a kind of Fabry-Perot cavity, such that the damping rate of that huge field is enough to cancel the incoming energy and provide perfect absorption. Is this the right picture? The reader could benefit from it if it is, and if it's not they would benefit from a clear explanation of where the energy is going under those conditions.

5) The authors are encouraged to criticize other approaches a bit more gently. Their results are interesting, and it will help to have a more even-handed treatment of the advantages and disadvantages of this kind of structure versus other approaches (i.e. optomechanics).

Reviewer #3:

Remarks to the Author:

The authors demonstrate how the non Hermitian nature of the interactions can be used to manipulate absorption and transparency in magnonics. New results in the paper is the condition for zero reflection for ϕ such that $\sin[2\phi]$ is nonzero and the experiments to confirm it. However earlier works [for example Mukhopadhyay et al, Phys. Rev A A 100, 013812 (2019)] have given analytical results for perfect zero reflection for a number of identical resonators coupled to waveguides. These papers also bring out the physics behind such a zero in reflection. The authors' result 5 is more general as it can also deal with cases when the resonators are nonidentical.

I find that there are several parts which are quite misleading and even wrong.

Theoretical description in terms of quantum Hamiltonian [1] is simply wrong- the authors must note that a quantum mechanical consistent description is in terms of the master equation for the density matrix of the system or in terms of the quantum Langevin Equations. Early workers, especially the developers of the quantum theory of laser and open system dynamics took great pains to bring it out. Hu's group has used such a description before but I am totally against it as it is wrong.

The abstract and text have discussions of coherent perfect absorption- the CPA refers to the interference of counterpropagating waves leading to zero transmission and reflection. Whereas the authors bring microwaves from one port only.

I have difficulty in understanding fig 4a- say the full black horizontal line $r_{sub1}(cal)$ —The authors previously discuss that r_{sub1} can be zero only for $\sin[2\phi]$ nonzero- but these values of ϕ are shown in the theory part. This is quite misleading.

The level attraction and repulsion discussion in lines 139 to 291 is well known and does not add anything new.

Thus in summary I do not recommend publication of the paper given that 1) the conditions for perfect zero reflection have been obtained earlier, 2) the theoretical description [eq 1] is incorrect, 3) paper has several misleading parts as described above.

Reply to Reviewer #1

The authors have presented a very interesting work on cavity magnonics where two YIG spheres are coupled to a same microwave transmission line resonator. Several things have been shown here including indirect coupling between the two samples, and level repulsion and level attraction. While the results are interesting, I do not feel this paper meets the criterial for Nature Communications:

Reply: We thank the Reviewer very much for evaluating our work to be “very interesting” and we are also grateful to the critical comments which are very much valuable for us to improve our manuscript. We have addressed each point of the report and improved our manuscript accordingly.

1. The results are interesting but there is a lot of superfluous language and terms that do not add to the discuss, instead they take away from the main findings. For instance, from the title with 'Transparency in Reflectionless' magnonics. I am unsure what this means. Are the authors quantifying transparency and reflection of microwaves (or the traveling waves) in their devices? Does not seem to be the case as they are working with the hybridisation. If they want to quantify the traveling wave behaviour, shouldn't an electromagnetic theory with the full calculation of these be needed?

Reply: We thank the Reviewer for pointing out our language problems. We have proofread and improved the whole manuscript, like removing the superfluous language and making it much clearer. We slightly changed the title from “Non-Hermitian Control between Absorption and Transparency in Reflectionless Magnonics” to “Non-Hermitian Control between Absorption and Transparency in Perfect Zero-Reflection Magnonics” so that the main experimental finding---Perfect Zero-Reflection (PZR)---becomes more explicit. In the improved Fig. 3, we added both the calculated Zero-Reflection (ZR) states and the hybridized eigenstates in the measured reflection spectra (Figs. 3a-c for S_{11} and Figs. 3g-i for S_{22}), and in Supplementary Note 3, we also provided the detail experimental mappings and the corresponding numerical results (FIGs. S3 and S4), so that the excellent consistency between our experiments and theory is clearly demonstrated. When the complex frequency of ZR ($\tilde{\omega}_{ZR}$) becomes purely real (circle marked in Fig. 2f (h)), the PZR occurs as a singularity in reflection, and it can be directly observed in experiment as an infinitely sharp dip shown in Figs. 3f (l). In addition, to quantify the travelling wave behaviour, we have added the full analytical description of reflection (S_{11} , S_{22}) and transmission (S_{21} , S_{12}) in Eqs. 3 and 4, more specifically, we have also added the new form of S-parameter under the PZR condition as shown in Eqs. 6a and 6b.

2. The authors results seems very similar to their own previous work. There is a vast literature now on dissipative coupling, which some of the authors are involved, and this seems to be an extension of that.

Reply: We thank the reviewer for drawing our attention to distinguish our results from our previous works. The main difference from early works is that our present work is the first study of zero-reflection (ZR) *in the complex frequency plane* of non-Hermitian cavity magnonics, including the specific perfect zero-reflection (PZR), a new singularity in reflection spectra, which is crucial for avoiding efficiency sacrifice and signal disturbances in device [e.g., Nat. Photonics, 7, 579 (2013)]. While our previous works, as reviewed, e.g., in JAP 127, 130901 (2020) / JAP 129,

201101 (2021) / Physics Reports 979, 1-61 (2022), have not addressed ZR or PZR. In fact, PZR is hard to be observed simultaneously in both S_{11} and S_{22} in our previous works, as briefly explained in the following:

In our previous works, the sample typically consists of one metal cavity resonator and one magnonic resonator which therefore have very different damping rates ($\kappa_c \gg \kappa_m$). Qualitatively, this *unbalanced* parameters results in a Fabry-Pérot (FP)-like system with very poor PF characteristics due to one mirror in this system is high lossy while the other mirror is not. While in our present work, the same type of resonators (two magnonic resonators) with *balanced* damping rates ($\kappa_{1,2}$ in the range of 1~4 MHz) ensure the good FP characteristics of our system. More quantitatively, Eq. 5 in this work provides a general requirement for the PZR to appear in a coupled two-resonators system. For simplicity and clarity, we consider a limit case with *unbalanced* parameters of $\kappa_c \gg \gamma_c \simeq \kappa_m \simeq \gamma_m$ like previous works. According to Eq. 5, in order to get meaningful Φ solutions satisfying ω_{PZR}^{\pm} , the frequency detuning between two resonators ($\Delta_H \equiv \omega_c - \omega_m$) has to be larger than $|\Delta_H| \geq \kappa_c \sqrt{\left(2 + \frac{\gamma_m}{\kappa_m}\right) \frac{\gamma_m}{\kappa_m}} \simeq \sqrt{3}\kappa_c$. This implies that the PZR will not appear in the coupled region ($\omega_c \simeq \omega_m$). Therefore, in previous experiments, even if the reflection spectra were acquired, the PZR cannot be observed within the limited measurement frequency range since κ_c is large.

In our revised manuscript, we have added the comparison between our present work and previous works, and we emphasize that the balanced parameters (all of κ_1 , κ_2 , γ_1 , γ_2 are in the same range of 1~4 MHz, about three orders smaller than the working frequency 3.5~5 GHz) are important for the observed PZR in S_{11} and S_{22} . Correspondingly, we have added a sentence on Page 5 Lines 165-171 as “**Second, different from previous works using two different types of resonators, namely, metal cavity resonator and magnonic resonator^{25,26,38}, the two same magnonic-type resonators in this work allows them more balanced in terms of the intrinsic and extrinsic damping rates ($\gamma_1, \gamma_2, \kappa_1, \kappa_2$). The balanced dampings, in particular $\kappa_1 \sim \kappa_2$, are important for establishing the balanced FP-like resonance since the reflection at the YIG 1(2) is proportional to the corresponding $\kappa_{1(2)}$ (see Supplementary Note 1). This balanced FP-like resonance allows the ZR condition can be observed near the coupling region...**”.

3. Related to 2, the authors do not contextualise their data in terms of previous literature. In fact, the work is presented as almost unrelated to previous mechanisms which does not seem to be the case.

Reply: We thank the Reviewer for this kind suggestion. In the revision, we have added the discussion of the relation with and the difference from previous mechanisms, particularly, the mode hybridization on Page 4, 5 Lines 134-171. For the ZR and PZR, we added a new reference (Ref. 30) which theoretically predicted a kind of PZR mechanism, and then, we discussed the main difference of our work to that work. The related discussions have been added on Page 7 Lines 230-233 “**Earlier work by Mukhopadhyay *et al*³⁰ proposed similar physics behind PZR by coupling a number of identical and lossless resonators to a common waveguide. Here, Eq. (5) provides a more general formulism for nonidentical and lossy resonators with traveling-photon-mediated indirect coupling**”.

4. Related to 3. In Figure 3, the results seems virtually identical to the 'level-merging' behavior

discussed by Boventre et al <https://doi.org/10.1103/PhysRevResearch.2.013154>. What is the relationship between a two port cavity system and this one. Seems to be deeply related but again, relationship to previous literature is missing.

Reply: We thank the Reviewer for drawing our attention to Boventre *et al*'s very interesting work. They utilized two excitations to drive the cavity mode and magnon mode respectively, the relative phase (ϕ) and amplitude ratio (δ_0) between the two driving fields are used as tuning knob to control the mode coupling ($g_{eff}\sqrt{1 + \delta_0 e^{i\phi}}$). In the limit of small magnon drive ($\delta_0 \rightarrow 0$), the real part of the coupling strength merges into a same one ($g_{eff}\sqrt{1 + \delta_0 e^{i\phi}} \rightarrow g_{eff}$), in level repulsion region (being independent of the relative phase ϕ). While in our work, since we use only one excitation, our device in some sense works in the limit of $\delta_0 \rightarrow 0$ of Boventre *et al*'s work. On the other hand, we utilize another degree of freedom of the *indirect* coupling between the two resonators. The propagating delay phase (Φ) arise from the large distance enables the regulation of coupling strength ($-i\Gamma e^{i\Phi}$) covering both level attraction and level repulsion regions, unlike the only remaining level repulsion in Boventre *et al*'s case when $\delta_0 \rightarrow 0$, $g_{eff}\sqrt{1 + \delta_0 e^{i\phi}} \rightarrow g_{eff}$. The tunability with Φ in our work can be ascribed to the fact that the direct coupling through modes overlapping (often leads to coherent coupling) has been removed completely due to the large spatial separation, and both the level repulsion (Fig. 2c) and level attraction (Fig. 2a) originate from the long-range indirect coupling, where the effective coherent term ($-i\Gamma e^{i\Phi}$ at $\Phi = (n + 1/2)\pi$) results in LR and the effective dissipative term ($-i\Gamma e^{i\Phi}$ at $\Phi = n\pi$) results in LA.

The second large difference of our work from Boventre *et al*'s work is about zero-reflection (ZR). In Boventre *et al*'s work, the reflection spectra (Fig. 2 in their paper) show only hybridized modes without any indication of the ZR. We surmise that the complex ZR frequency ($\tilde{\omega}_{ZR}$) in their work has very large imaginary part, as a result, the measured spectral region is far from the PZR region ($|\text{Im}(\tilde{\omega}_{ZR})| \gg 0$) and hence less affected by ZR line. In fact, stringent conditions between variables like the frequency detuning ($\Delta_H \equiv \omega_1 - \omega_2$) and phase (Φ) has to be satisfied (Eq. 5 in our work) in order to realize the experimentally observable PZR condition. In our work, the reflection spectra show not only the hybridized modes, but also the strong indication of ZR and PZR, as we demonstrated in Fig. 3 (main text) and FIGs. S3, 4 (Supplementary Note 3).

In the revised manuscript, we have added the citation of Boventre *et al*'s work (Ref. 40), and also discuss the relation to our work on Pages 4, 5 Lines 134-171 **“The mode hybridization appears to be similar to previous works³⁸⁻⁴⁰, covering both level attraction (LA) and level repulsion (LR) regions...On the other hand, our work distinguishes from earlier works in the two aspects: First, traveling photon bath... Second...balanced FP-like resonance allows the ZR condition can be observed near the coupling region...”**.

5. Also related to 2. How does level attraction related to the Lenz effect as discussed by Harder et al <https://doi.org/10.1103/PhysRevLett.121.137203>?

Reply: We thank the Reviewer for this very nice question. Harder *et al*'s work provided a classical and phenomenological electrodynamic picture with Lenz effect to explain the observed level attraction and associated dissipative coupling between the metal cavity and the *embedded*

magnonic resonator. Subsequent research [PRL 123, 127202 (2019)] has revealed that additional (non-resonant) microwave mode has been involved in the dissipative coupling, which functions as a common reservoir (dissipative channel) to both cavity and magnon. Their cooperative interaction with the common reservoir induces an effective dissipative coupling between the two modes. Our present work discloses in a decent way the important role of the common reservoir (i.e., transmission line) in producing the dissipative coupling and level attraction: As shown in Eqs. 1 and 2, pure dissipative coupling takes place when $\Phi = 2n\pi$ (in-phase) or $\Phi = (2n + 1)\pi$ (180-degree out-of-phase). This implies that their cooperative interaction with the common reservoir tends to synchrotronize two resonators (equalize their real part of the eigenfrequency) when dissipative coupling takes place, being consistent with the explanation in Fig. 5 of JAP 129, 201101 (2021). In terms of Lenz effect in present work, first we need to separately consider for each magnon resonator interacting with the transmission line. The dynamic magnetization m_i ($i=1,2$) in magnon resonator affects the local rf current j_i ($i=1,2$) in the transmission line through Faraday's law. In turn, the local rf current j_i produces rf magnetic field and back-acts to dynamic magnetization m_i (through both Ampere's law and Lenz effect). This interaction and backaction with the transmission line are reciprocal at each magnonic resonator m_i . Then when we consider the effective mutual induction interaction ($m_1 \Leftrightarrow j_2$ and $m_2 \Leftrightarrow j_1$) due to the indirect coupling, we can notice that Lenz effect contributes effectively to the new eigenvectors $[m_1, \pm m_2]$ if we regard magnon resonators as magnetic coils with long-range inductive interaction: Lenz effect dominates over Ampere's law for the eigenvector of $[m_1, -m_2]$ while Ampere's law dominates over Lenz effect for the eigenvector of $[m_1, m_2]$.

Finally, it is my feeling that the authors would need to address points above before this paper is considered for publication.

Reply: We thank the Reviewer again for careful reviewing and kind suggestions. We hope that we have properly addressed each point of the report and have improved the manuscript to a satisfying level.

Reply to Reviewer #2

The results presented here are very interesting, and show evidence of tunable control of a system set up with coherent magnonic resonators (based on YIG spheres) to yield interfering systems whose reflection, transmission and absorption features are tunable. The authors tune the system to one of very low reflectivity and then show that it is possible to control whether the system is transmissive or absorbing by changing the phase between the two oscillators (using a tuning field).

The paper itself is more difficult to read than it should be, due to the extensive use of specific jargon, the presentation of some statements that are confusing, a lack of clarity about when experimental and when theoretical results are being discussed, and missing physical pictures to help the reader understand intuitively the features. If these are addressed this paper is likely a significant contribution to the fundamental exploration of microwave transmission, absorption, and reflection control, and could be appropriate for Nature Communications.

Reply: We thank the Reviewer very much for evaluating our work to be "very interesting" and "...likely a significant contribution to the fundamental exploration...". We are also grateful for the

critical comments to the readability of our manuscript, to which we fully agree. So, in the revised manuscript, we have tried our best to improve the readability by, e.g., adding intuitive physical pictures of the PZR (Fig. 1d) and schematic configuration on different hybridized states (top cartoons in Fig. 2). We hope that the improved manuscript is easier to understand and can be published for Nature Communications. Detail replies are shown point-by-point as follow:

1. PA, PT, and PZR are unfortunately confusing, since PZR means perfect zero reflection (hence reflection = 0) whereas PA and PT mean perfect absorption and perfect transmission (i.e. =1 not =0). The terms are also unfortunate since nothing is ever "perfect". Even A+T+R is not precisely equal to 1, since there is potentially scattering that would occur out of the waveguide.

Reply: We thank the Reviewer very much for this comment and we are sorry for our unclear description and confusing terminology. After reconsidering the consistency and clarity throughout our whole manuscript, we prefer to change “perfect absorption (PA)” to “full absorption (FA)”, and “perfect transparency (PT)” to “full transmission (FT)”, in which “full” means the amplitude approaches 1. Also, “Perfect Zero Reflection (PZR)” is changed slightly to “Perfect Zero-Reflection (PZR)” (following PRB 93, 104202 (2016) although the topic is different), which means the solution of Zero-Reflection (ZR) *in the complex frequency plane* becomes purely real with zero imaginary part, i.e., $\text{Im}(\tilde{\omega}_{\text{ZR}}) = 0$. This also implies that the general complex solution of ZR with *non-zero* imaginary part ($\text{Im}(\tilde{\omega}_{\text{ZR}}) \neq 0$) could be called *Imperfect Zero-Reflection* because, in this case, reflection becomes zero only for a complex frequency and the real experimental measurement is projected to the real frequency axis. Therefore, it will not give zero-reflection but just show a reflection dip with rather large bandwidth depending on the amplitude of $\text{Im}(\tilde{\omega}_{\text{ZR}})$. This can be seen clearly in, e.g., Fig. 3d (near the PZR). For the PZR, the reflection reaches strictly zero theoretically and therefore the PZR appears as a singularity with infinitely narrow linewidth (Fig. 3f) and infinite group delay discontinuity (Fig. 3e). We hence hope that “Perfect” can be acceptable to be used in the PZR case. On the other hand, we fully agree that nothing can be physically perfect, particularly in real experiments (depending on e.g., the accuracy of the setup). Therefore, we pay special attention to give real numbers in dB when we describe the PZR dip in our revised manuscript. Also, to be more accurate scientifically. We add “nearly” before full absorption (as NFA) and full transmission (as NFT) throughout the manuscript as the full absorption/transmission will be approached infinitely but never be reached strictly.

2. a statement like "perfect zero reflection is a prerequisite" is troubling since again there is always going to be some imperfect tuning and some reflection. The authors should provide bounds for these features rather than saying "negligible" or "zero" so the readers know to what level of accuracy these statements are experimentally valid.

Reply: We thank the Reviewer for this comment. We have improved the description and provided the experimental bounds in our work, which is -67 dB at the PZR frequency with a very narrow bandwidth of ~0.1 MHz. Accordingly, the statement “**ultra-sharp dip about 67 dB is observed with the measuring accuracy of ~0.1 MHz (Fig. 3f or Fig. 3l)**” has been added on Page 7 Lines 229-230, and “**Beside the ultra-sharp dip (~67 dB with ~0.1 MHz measuring accuracy) observed in the reflection spectrum (Fig. 3f (3l))...**” has been added on Page 7 Line 256.

3. it is often difficult to tell what pieces are theoretical and what pieces are experimental. Many figures are presented without this clarity (i.e. fig. 2) and only later is it apparent that this is entirely theoretical. The authors should be upfront about which results are theoretical and which experimental.

Reply: We thank the Reviewer to remind us to distinguish the theoretical pieces and experimental pieces more clearly. We have followed this suggestion and improved the manuscript. In Fig. 2, we have changed legend “**The evolution between level attraction and level repulsion**” to “**The theoretical calculations of zero-reflection (ZR) condition and mode hybridization**” to stress this part is the theoretical prediction. In Fig. 3, the legend “**Perfect zero-reflection condition**” has been changed to “**Perfect zero-reflection observed in experiment**” in order to stress it is the experimental verification. Also, the statement “**Theoretical prediction and experimental verification...**” has been added in the legend of Fig. 4.

4. a key insight that would help the reader is - for perfect absorption, where is the energy going? the authors have emphasized the negligible loss of the YIG spheres, but is that loss what is driving the absorption? I wonder if the effect of the structure is to maintain a huge microwave field in a kind of Fabry-Perot cavity, such that the damping rate of that huge field is enough to cancel the incoming energy and provide perfect absorption. Is this the right picture? The reader could benefit from it if it is, and if it's not they would benefit from a clear explanation of where the energy is going under those conditions.

Reply: We thank the Reviewer for this very nice and inspiring question. As Reviewer expected, the two magnonic resonators configured in our system can be regarded as the paired mirrors of a Fabry-Pérot (FP) cavity. In our revised manuscript, we have added one supplementary section (**Supplementary Note 1. (3)**) to explain the Fabry-Pérot (FP)-like resonant mechanism. Remarkably, *in the complex frequency plane*, the assumption of *lossless* round-trip propagation of trapped photons inbetween two YIGs (mirrors) will give exactly the same eigenfunction as Eq.(1). This suggests that our two-magnon system is indeed a kind of Fabry-Perot cavity which can maintain a huge microwave field inside the cavity. Note that one prominent difference of our FP-like mechanism from the conventional FP resonance is the “mirrors (i.e., YIG spheres)” in our work are dispersive and hence the phase shift by two YIG spheres has to be taken into account (see Eqs. S30-S33). Since the actual experiment is projected to *the real frequency axis*, the round-trip propagation of trapped photons becomes lossy. In the nearly full absorption (NFA) region, the intrinsic damping (γ_1, γ_2) of two YIGs becomes the dominant (if not only) energy dissipation channel when the leakage at both ports (i.e., reflection and transmission) are minimized to (nearly) zero. (Our theoretical model neglects any other loss like the transmission line loss and the calculation result with this model agrees surprisingly with experiments which therefore suggests that other loss mechanisms are indeed negligibly small.) This condition can be approached when (i) Eq.(5) is satisfied to suppress reflection (i.e., PZR); (ii) the frequency detuning is small enough so that absorption at two YIGs are maximized and hence $S_{21} \rightarrow 0$. In the limit of small frequency detuning ($\omega_{PZR} \cong \omega_{1(2)}$), the absorption by YIG1 (YIG2) can be derived

to be $\frac{2^{\gamma_{1(2)}}}{\kappa_{1(2)}} \frac{1}{\left(\frac{\gamma_{1(2)}}{\kappa_{1(2)}} + 1\right)^2}$ for each round-trip of photon FP propagation, which can reach ~50% if $\gamma_{1(2)}$

matches $\kappa_{1(2)}$, i.e., $\gamma_{1(2)} = \kappa_{1(2)}$. This explains why the loss of YIG is still dominating the total

absorption although $\gamma_{1(2)}$ itself is small (compared to the working frequency ~ 4 GHz).

In the revised manuscript, we have added the description about the energy dissipation in nearly full absorption region and also the FP-like mechanism. On Pages 8, 9 Lines 291-298, we add “As shown schematically in Fig. 1d, in the nearly full absorption (NFA) region ($\omega_1 \cong \omega_2$), FP-like mechanism for photons trapped between the two resonators assists the magnon exchange between two YIG spheres, and eventually, most of the microwave energy is dissipated through the spin precessions in the two YIG spheres despite the intrinsic losses of two magnon modes are rather small (~ 1 MHz). While in the nearly full transmission (NFT) region, the large frequency detuning ($\omega_1 \neq \omega_2$) makes two magnon modes are nearly uncoupled with each other at the PZR frequency, so that little microwave energy can be absorbed by YIG spheres since $\omega_{PZR} \neq \omega_1$ and $\omega_{PZR} \neq \omega_2$.”.

Duplicate of Supplementary Note 1. (3) Fabry-Pérot (FP)-like resonant mechanism. As shown in FIG. S2, considering microwaves are loaded from the left side of resonator a , and multiple reflect at two resonators.

Duplicate of Figure S2 | Schematic diagram of wave propagation in two-magnon system.

For the isolated mode a , the reflection and transmission are in the forms,

$$S_{11}^a(\omega) = S_{22}^a(\omega) = -\frac{2i\omega\kappa_1}{\omega^2 - \omega_1^2 + 2i\omega(\gamma_1 + \kappa_1)} \quad (\text{S26})$$

$$S_{21}^a(\omega) = S_{12}^a(\omega) = 1 - \frac{2i\omega\kappa_1}{\omega^2 - \omega_1^2 + 2i\omega(\gamma_1 + \kappa_1)} \quad (\text{S27})$$

For the isolated mode b , the reflection and transmission are in the forms,

$$S_{11}^b(\omega) = S_{22}^b(\omega) = -\frac{2i\omega\kappa_2}{\omega^2 - \omega_2^2 + 2i\omega(\gamma_2 + \kappa_2)} \quad (\text{S28})$$

$$S_{21}^b(\omega) = S_{12}^b(\omega) = 1 - \frac{2i\omega\kappa_2}{\omega^2 - \omega_2^2 + 2i\omega(\gamma_2 + \kappa_2)} \quad (\text{S29})$$

Considering the round-way behavior between two modes (interaction loop between modes a and b as shown in FIG. S2) and defining,

$$w = S_{22}^a(\omega)e^{i\Phi}S_{11}^b(\omega)e^{i\Phi} \quad (\text{S30})$$

Substituting Eqs. (S26) and (S28) into Eq. (S30), it can be written as,

$$w = \frac{-2i\omega\kappa_1}{\omega^2 - \omega_1^2 + 2i\omega(\gamma_1 + \kappa_1)} \frac{-2i\omega\kappa_2}{\omega^2 - \omega_2^2 + 2i\omega(\gamma_2 + \kappa_2)} e^{2i\Phi} \quad (\text{S31})$$

Setting $w \equiv 1$ in Eq. (S31) and taking the approximation $\omega^2 - \omega_1^2 \sim 2\omega(\omega - \omega_1)$ and $\omega^2 - \omega_2^2 \sim 2\omega(\omega - \omega_2)$ (near the coupling region), we obtain,

$$\left[\frac{i\kappa_1}{\omega - \omega_1 + i(\gamma_1 + \kappa_1)} \right] \left[\frac{i\kappa_2}{\omega - \omega_2 + i(\gamma_2 + \kappa_2)} \right] e^{2i\Phi} = 1 \quad (\text{S32})$$

This is exactly the eigen-equation as follow,

$$[\omega - \omega_1 + i(\gamma_1 + \kappa_1)][\omega - \omega_2 + i(\gamma_2 + \kappa_2)] + \kappa_1\kappa_2 e^{2i\Phi} = 0 \quad (\text{S33})$$

The eigenvalues ($\tilde{\omega}_\pm$) in Eq. (2) of the main text is the solution of above eigen-equation, which is also consistent with the theory we discussed in Supplementary Note 1 (1) and (2).

5. The authors are encouraged to criticize other approaches a bit more gently. Their results are interesting, and it will help to have a more even-handed treatment of the advantages and disadvantages of this kind of structure versus other approaches (i.e. optomechanics).

Reply: We thank the Reviewer for kind suggestion and we apologize for the inappropriate comment to other approaches, which we did not really mean it. We have thoroughly proofread the whole manuscript and adjusted the comment to be soft and justified. For example, we have improved the descriptions in abstract which are highlighted by blue, and added citations and descriptions about optomechanics on Page 2 Line 53. We hope that the improved manuscript will be acceptable to all readers from relevant but different fields.

Reply to Reviewer #3

The authors demonstrate how the non Hermitian nature of the interactions can be used to manipulate absorption and transparency in magnonics. New results in the paper is the condition for zero reflection for phi such that $\sin[2\phi]$ is nonzero and the experiments to confirm it.

Reply: We thank the Reviewer for careful reviewing and for evaluating our result of ZR with nonzero $\sin(2\Phi)$ to be new. We are also grateful to the critical comments which inspired us to improve our manuscript significantly. Detail replies are shown point-by-point as follow:

However earlier works [for example Mukhopadhyay et al, Phys. Rev A 100, 013812 (2019)] have given analytical results for perfect zero reflection for a number of identical resonators coupled to waveguides. These papers also bring out the physics behind such a zero in reflection. The authors' result 5 is more general as it can also deal with cases when the resonators are nonidentical.

Reply: We thank the Reviewer for drawing our attention to earlier works and also for evaluating our result (Eq. 5) to be more general for non-identical resonators. We have carefully studied earlier works including Mukhopadhyay *et al*'s and also the relation with our work. We found that perfect zero-reflection (PZR) in Mukhopadhyay *et al*'s work is based on stringent theoretical assumptions with not only *identical* resonators but also *lossless* resonators. Namely, these assumptions include: (i) identical resonators as Reviewer pointed out, i.e., $\omega_0 \equiv \omega_1 = \omega_2$; (ii) identical coupling to the common waveguide, i.e., $\Gamma \equiv \kappa_1 = \kappa_2$; and (iii) lossless resonators, i.e., $\gamma_0 \equiv \gamma_1 = \gamma_2 = 0$. Analytically, we can derive for this identical and lossy (lossless) two-resonator system that zero reflection (ZR) requires $\left(\tan \Phi + \frac{\Delta}{\Gamma}\right)^2 + \left(\frac{\gamma_0}{\Gamma}\right)^2 = 0$, where $\Delta \equiv \omega - \omega_0$ is the frequency

detuning. It can be seen that both $\gamma_0 = 0$ and $\tan \Phi + \frac{\Delta}{\Gamma} = 0$ have to be satisfied simultaneously,

as shown in Fig.2 in Mukhopadhyay *et al*'s paper (we have reproduced those curves although not shown here). This implies that, for the practical resonators with nonzero intrinsic dampings ($\gamma_0 \neq$

0), the coupled identical-resonator model will not give the singular PZR behavior. In our work, however, we demonstrate that PZR singularities exist generally for the practical lossy resonators when the non-identical degree of freedom is introduced (e.g., $\omega_1 \neq \omega_2$) and we experimentally verified the existence of PZR by fine adjustment of parameter space with both the frequency detuning ($\Delta_H \equiv \omega_1 - \omega_2$) and propagation delay phase (Φ). Besides, the transmission/absorption can be controlled accordingly, which is beyond earlier works.

In our revised manuscript, we have added more relevant references (Refs. 30, 31) including Mukhopadhyay *et al*'s work (Ref. 30), and discuss briefly the difference and generality of our work, as **“Earlier work by Mukhopadhyay *et al*³⁰ proposed similar physics behind PZR by coupling a number of identical and lossless resonators to a common waveguide. Here, Eq. (5) provides a more general formulism for nonidentical and lossy resonators with traveling-photon-mediated indirect coupling.”** on Page 7 Lines 230-233.

I find that there are several parts which are quite misleading and even wrong.

Theoretical description in terms of quantum Hamiltonian [1] is simply wrong- the authors must note that a quantum mechanical consistent description is in terms of the master equation for the density matrix of the system or in terms of the quantum Langevin Equations. Early workers, especially the developers of the quantum theory of laser and open system dynamics took great pains to bring it out. Hu's group has used such a description before but I am totally against it as it is wrong.

Reply: We thank the Reviewer very much for this critical comment which may arises from our unclear description in our original manuscript. Firstly, we should emphasize that the Hamiltonian described in Eq. 1 is an *effective* non-Hermitian of our open system. This “effective Hamiltonian” language has been widely used in a broad range of area including, e.g., in atomic system [new Ref. 37, Nat. Phys 12, 1139 (2016)], cavity quantum electrodynamics [new Ref. 34, Nature 569, 692 (2019)], optomechanical systems [new Ref. 35, Phys. Rev. X 5, 021025 (2015)] and so on. Secondly, the validity of the Hamiltonian is supported independently by the different approach, namely ABCD matrix method (Ref. 44 and Supplementary Note 1), and we mentioned in Line 205 on page 8 of our original manuscript that **“...the eigenstates in Eq. (2) correspond to zero denominator in Eqs. (3) and (4) or divergent S11 and S21.”** to show the consistency between effective Hamiltonian (Eq. 1) and ABCD matrix method. However, we realized that this statement is rather too mathematical and physically not transparent enough to readers. We therefore consider to improve our description and prove the validity of the effective Hamiltonian in three different aspects:

(i) We derive the effective Hamiltonian of our system using the Feshbach projection approach. This Feshbach projection approach, though different from the master equation or quantum Langevin Equations, has been widely used in non-Hermitian physics field [new Ref. 36, Advances in Physics 69, 249-435 (2020), Chapter 4.1] and provides reliable result.

(ii) We improve the ABCD matrix description and emphasize that poles of S-parameters in complex frequency plane correspond to the eigenvalues (see, e.g., Ref. 28, Science 373, 1261 (2021)) and therefore zero-denominator (divergence) of S-parameters corresponds to the eigen-equation.

(i) and (ii) are independent but give consistent results, making us confident about the validity of the effective Hamiltonian (Eq. 1).

(iii) To further confirm the validity of the effective Hamiltonian, we directly compare the calculated eigenvalues (real part) with the experimentally measured spectra, as shown in Fig.3. Excellent agreement between the theory (dotted curves) and experiments (dark region in color plot) can be clearly seen.

Combining (i)-(iii), we are convinced that our effective Hamiltonian shown in Eq. 1 is correct. Besides, we understand that Reviewer's concern may arise from the earlier mathematical agreement but physically not transparent Hamiltonian description in some of our previous works (e.g., not physically well-defined Φ in the off-diagonal term of the effective Hamiltonian in previous work), partly due to the complicated experimental configuration. However, we stress that our present work is free from such an issue as the configuration is rather simple enough with all parameters (e.g., the microwave propagating phase Φ) being well-defined both mathematically and physically. We therefore hope that the present work being more physically transparent and convincing can be accepted for publication. To elaborate the equivalent Hamiltonian more rigorously, we improve some statements in the main text Page 4 Line 122-125, as “**Under the rotating-wave approximation, the effective non-Hermitian Hamiltonian of the subsystem can be created by using Feshbach projection approach³⁶ (see Supplementary Note 1 for details), $\mathcal{H}_{eff} = \hbar\tilde{\omega}_1\hat{a}^\dagger\hat{a} + \hbar\tilde{\omega}_2\hat{b}^\dagger\hat{b} - \hbar(i\Gamma e^{i\Phi})(\hat{a}^\dagger\hat{b} + \hat{b}^\dagger\hat{a})$ ”.**

In the following, we describe the above three aspects (i)-(iii) in more detail, which we have also included in our revised manuscript and supplementary.

(I) Derivation of the effective non-Hermitian Hamiltonian by Feshbach projection approach

Under the rotating-wave approximation, the Hamiltonian of the whole system is,

$$H/\hbar = \tilde{\omega}_1\hat{a}^\dagger\hat{a} + \tilde{\omega}_2\hat{b}^\dagger\hat{b} + \sum_k \omega_k p_k^\dagger p_k + \sum_k \omega_k q_k^\dagger q_k + \sum_k g_1 a_1^\dagger(p_k+q_k) + \sum_k g_2 e^{i\Phi} a_2^\dagger(p_k+q_k) + h.c. \quad (R1)$$

where $\hat{a}^\dagger(\hat{a})$ and $\hat{b}^\dagger(\hat{b})$ represent the creation (annihilation) operators of the two magnon modes, $\omega_{1,2} - i\gamma_{1,2}$ are their complex frequencies, with the real and imaginary parts representing the resonant frequency and intrinsic damping rates, respectively. $p_k^\dagger(p_k)$ and $q_k^\dagger(q_k)$ are the creation (annihilation) operators of the rightward and leftward traveling photon modes, respectively, which follow the commutation relations $[p_{k'}^\dagger, p_{k'}] = \delta(k - k')$ and $[q_{k'}^\dagger, q_{k'}] = \delta(k - k')$. The second row of Eq. (R1) represents the interaction between the traveling photons and two magnon modes. $g_{1,2}$ are the coupling strength between magnon modes and traveling photon modes. Φ is the propagating phase between two magnon modes.

We only focus on the subsystem which is made of two magnon modes and embedded in traveling photon bath. To characterize the time evolution of this subsystem, we use Feshbach projection approach to create an effective non-Hermitian Hamiltonian of the subsystem, which can be expressed in these three terms,

$$\mathcal{H}_{eff}(\omega) = H_0 + \Delta(\omega) - \frac{i}{2}\Gamma(\omega) \quad (R2)$$

where the H_0 is the subspace of the H and can be written in matrix form,

$$H_0 = \begin{bmatrix} \omega_1 - i\gamma_1 & 0 \\ 0 & \omega_2 - i\gamma_2 \end{bmatrix} \quad (\text{R3})$$

the zero off-diagonal term means that there is no direct coupling between two magnon mode. The correction term $\Delta(\omega)$ and $\Gamma(\omega)$ represent the energy shift and decay due to indirect coupling. Before obtaining these two terms, we introduce the operator $A(\omega)$ to characterize the system-environmental coupling,

$$A(\omega) = \begin{bmatrix} g_1 & g_2 e^{-i\Phi} \\ g_1 & g_2 e^{i\Phi} \end{bmatrix} \quad (\text{R4})$$

the matrix element $A(\omega)_{nm} = \langle \phi_n | H | \psi_m \rangle$. Where the $\psi_{m=1,2}$ are the wave function of the m-th magnon mode, $\phi_{n=1,2}$ are the wave function of the rightward ($n = 1$) and leftward ($n = 2$) traveling photon modes, respectively.

Then $\Delta(\omega)$ and $\Gamma(\omega)$ can be expressed by A matrix.

$$\Delta(\omega) = \text{PV} \left(\int \frac{A^\dagger(\omega') A(\omega')}{\omega - \omega'} d\omega' \right) = \begin{bmatrix} 0 & \Gamma \sin\Phi \\ \Gamma \sin\Phi & 0 \end{bmatrix} \quad (\text{R5})$$

$$\Gamma(\omega) = 2\pi A^\dagger(\omega) A(\omega) = \begin{bmatrix} 2\kappa_1 & 2\Gamma \cos\Phi \\ 2\Gamma \cos\Phi & 2\kappa_2 \end{bmatrix} \quad (\text{R6})$$

where $g_{1,2} = \sqrt{\kappa_{1,2}/2\pi}$, $\Gamma = \sqrt{\kappa_1 \kappa_2}$, with $\kappa_{1,2}$ are the extrinsic damping rates of the two magnon modes.

The derivation of the $\Delta(\omega)$ uses the following fact,

$$\text{PV} \left(\int \frac{e^{i\omega't}}{\omega - \omega'} d\omega' \right) = -i\pi \text{sgn}(t) e^{i\omega t} \quad (\text{R7})$$

and $\Phi \propto \omega$.

Substituting Eqs. (R5) and (R6) into Eq. (R2), we obtain the effective non-Hermitian Hamiltonian,

$$H_{eff}(\omega) = \begin{bmatrix} \omega_1 - i(\gamma_1 + \kappa_1) & -i\Gamma e^{i\Phi} \\ -i\Gamma e^{i\Phi} & \omega_2 - i(\gamma_2 + \kappa_2) \end{bmatrix} \quad (\text{R8})$$

[Note: Here we notice a minor error of Eq. (1) in our original manuscript, as the dissipation is defined as negative imaginary part ($-i$), so that the sign before $\hbar(i\Gamma e^{i\Phi})$ should be changed as “ $-$ ”. On the other hand, this sign change will not affect all the results in this manuscript since the off-diagonal term is squared in Eq. (2). We thank the Reviewer for this questioning which encouraged us to double-check and correct this sign change and make the formula more accurate physically.]

The S matrix is given by,

$$S(\omega) = C \left\{ I - 2\pi i A(\omega) \frac{1}{\omega - H_{eff}(\omega)} A^\dagger(\omega) \right\} \quad (\text{R9})$$

where $C = \begin{pmatrix} 0 & 1 \\ 1 & 0 \end{pmatrix}$ and I is identity matrix.

The matrix element of the S matrix is equivalent to Eqs. (R23)-(R26) that we would present below.

(II) S-parameter analysis by ABCD matrix method

(i) One magnon coupled waveguide.

Figure R1 | (a) Schematic diagram of the waveguide magnonic device when a single YIG sphere couples to a transmission line. (b) Equivalent circuit of the single-magnon system.

A single YIG sphere (YIG 1) coupled to transmission line (Fig. R1 (a)) can be regarded as a resonant circuit with resistance R_1 , inductance L_1 and capacitance C_1 connected in series as shown in Fig. R1 (b). For this two-port network, the ABCD matrix is $\begin{bmatrix} A & B \\ C & D \end{bmatrix} = \begin{bmatrix} 1 & 0 \\ Z_1^{-1} & 1 \end{bmatrix}$, where $Z_1 = R_1 - i\omega L_1 - 1/i\omega C_1$ are the equivalent impedance. Defining resonant frequency $\omega_1 = 1/\sqrt{L_1 C_1}$, intrinsic damping rate $\gamma_1 = R_1/2L_1$ and extrinsic damping rate $\kappa_1 = Z_0/4L_1$ ($Z_0 = 50 \Omega$ is the characteristic impedance contributed from transmission line), so that $Z_1 = -iZ_0/4\kappa_1\omega(\omega^2 - \omega_1^2 + 2i\omega\gamma_1)$. Then, S-parameter of the system can be obtained from ABCD matrix. For instance, the transmission is derived as,

$$S_{21}(\omega) = \frac{2}{A + B/Z_0 + CZ_0 + D} = 1 - \frac{Z_0}{2Z_1 + Z_0} = 1 - \frac{2i\kappa_1\omega}{\omega^2 - \omega_1^2 + 2i\omega(\gamma_1 + \kappa_1)} \quad (\text{R10})$$

Considering frequency near the coupling region, $\omega^2 - \omega_1^2 \sim 2\omega(\omega - \omega_1)$,

$$S_{21}(\omega) = 1 - \frac{i\kappa_1}{\omega - \omega_1 + i(\gamma_1 + \kappa_1)} \quad (\text{R11})$$

which is the common form of transmission spectra for a single resonator coupling to waveguide.

(ii) Two magnons coupled waveguide.

Figure R2 | (a) Schematic diagram of the device with two YIG spheres couples to waveguide. (b) The ideal topology of the system consists of two resonant circuits (two YIG spheres) and a transmission line, M_1 , M_t , M_2 represent the cascade matrices of resonator 1, transmission line, and resonator 2, respectively. (c) Equivalent circuit of the coupled system.

As shown in Fig. R2 (a), two resonators (two YIG spheres) side-couple to a common transmission line. The RLC circuit model as shown in Fig. R2 (c) is developed to characterize this system, two YIG spheres act as two resonant circuits with self resistance $R_{1,2}$, inductance $L_{1,2}$ and capacitance $C_{1,2}$ connected in series, the subscript 1, 2 represent YIG 1 and YIG 2, respectively, and they are connected in parallel with the transmission line contributing characteristic impedance $Z_0 = 50 \Omega$, as well as the electrical length $\Phi = kl = 2\pi l/\lambda$. The complex impedance of two resonant circuits can be written as $Z_1 = R_1 - i\omega L_1 - 1/i\omega C_1$ and

$Z_2 = R_2 - i\omega L_2 - 1/i\omega C_2$. Because the feature dimensions of our device are much smaller than the wavelength of the microwaves, the scattering properties can be modeled by the cascade matrices M_1 , M_t , M_2 as shown in Fig. R2 (b), here,

$$M_1 = \begin{bmatrix} 1 & 0 \\ Z_1^{-1} & 1 \end{bmatrix}, \quad M_t = \begin{bmatrix} \cos\Phi & -iZ_0\sin\Phi \\ -iZ_0^{-1}\sin\Phi & \cos\Phi \end{bmatrix}, \quad M_2 = \begin{bmatrix} 1 & 0 \\ Z_2^{-1} & 1 \end{bmatrix} \quad (\text{R12})$$

The ABCD matrix for the whole system can be calculated by multiplying the matrices of the individual two-port element,

$$\begin{bmatrix} A & B \\ C & D \end{bmatrix} = M_1 M_t M_2 = \begin{bmatrix} \cos\Phi - i\frac{Z_0}{Z_2}\sin\Phi & -iZ_0\sin\Phi \\ \left(\frac{1}{Z_1} + \frac{1}{Z_2}\right)\cos\Phi - i\left(\frac{Z_0}{Z_1 Z_2} + \frac{1}{Z_0}\right)\sin\Phi & \cos\Phi - i\frac{Z_0}{Z_1}\sin\Phi \end{bmatrix} \quad (\text{R13})$$

According to this ABCD matrix, the transmission and reflection spectra can be derived by,

$$S_{11}(\omega) = \frac{A + B/Z_0 - CZ_0 - D}{A + B/Z_0 + CZ_0 + D} \quad (\text{R14})$$

$$S_{21}(\omega) = \frac{2}{A + B/Z_0 + CZ_0 + D} \quad (\text{R15})$$

$$S_{12}(\omega) = \frac{2(AD - BC)}{A + B/Z_0 + CZ_0 + D} \quad (\text{R16})$$

$$S_{22}(\omega) = \frac{-A + B/Z_0 - CZ_0 + D}{A + B/Z_0 + CZ_0 + D} \quad (\text{R17})$$

where $S_{11}(\omega)$ and $S_{21}(\omega)$ represent the reflection and transmission when microwaves are input from port 1 of the waveguide, $S_{22}(\omega)$ and $S_{12}(\omega)$ represent the case when microwaves are input from port 2. As an example, we display the derivation of $S_{21}(\omega)$ in detail as follow,

Substituting ABCD of Eq. (R13) into Eq. (R15),

$$S_{21}(\omega) = \frac{2Z_1 Z_2 e^{i\Phi}}{(2Z_1 Z_2 + Z_0 Z_1 + Z_0 Z_2) - iZ_0^2 \sin\Phi e^{i\Phi}} \quad (\text{R18})$$

Substituting $Z_1 = -iZ_0/4\kappa_1\omega(\omega^2 - \omega_1^2 + 2i\omega\gamma_1)$ and $Z_2 = -iZ_0/4\kappa_2\omega(\omega^2 - \omega_2^2 + 2i\omega\gamma_2)$ into Eq. (R15),

$$S_{21}(\omega) = \frac{(\omega^2 - \omega_1^2 + 2i\omega\gamma_1)(\omega^2 - \omega_2^2 + 2i\omega\gamma_2)e^{i\Phi}}{[\omega^2 - \omega_1^2 + 2i\omega(\gamma_1 + \kappa_1)][\omega^2 - \omega_2^2 + 2i\omega(\gamma_2 + \kappa_2)] + 4\omega^2\kappa_1\kappa_2 e^{2i\Phi}} \quad (\text{R19})$$

Similarly, the other three S-parameters are derived as,

$$S_{11}(\omega) = -2i\omega \frac{\kappa_1[\omega^2 - \omega_2^2 + 2i\omega(\gamma_2 + \kappa_2)] + \kappa_2 e^{2i\Phi}[\omega^2 - \omega_1^2 + 2i\omega(\gamma_1 - \kappa_1)]}{[\omega^2 - \omega_1^2 + 2i\omega(\gamma_1 + \kappa_1)][\omega^2 - \omega_2^2 + 2i\omega(\gamma_2 + \kappa_2)] + 4\omega^2\kappa_1\kappa_2 e^{2i\Phi}} \quad (\text{R20})$$

$$S_{12}(\omega) = S_{21}(\omega) \quad (\text{R21})$$

$$S_{22}(\omega) = -2i\omega \frac{\kappa_1 e^{2i\Phi}[\omega^2 - \omega_2^2 + 2i\omega(\gamma_2 - \kappa_2)] + \kappa_2[\omega^2 - \omega_1^2 + 2i\omega(\gamma_1 + \kappa_1)]}{[\omega^2 - \omega_1^2 + 2i\omega(\gamma_1 + \kappa_1)][\omega^2 - \omega_2^2 + 2i\omega(\gamma_2 + \kappa_2)] + 4\omega^2\kappa_1\kappa_2 e^{2i\Phi}} \quad (\text{R22})$$

Considering frequency near the coupling region, $\omega^2 - \omega_{1,2}^2 \sim 2\omega(\omega - \omega_{1,2})$, which is the classical version of the rotating-wave approximation ignoring the term of high-frequency oscillation, then, these four S-parameters are written as,

$$S_{11}(\omega) = -i \frac{\kappa_1[\Delta_2 + i(\gamma_2 + \kappa_2)] + \kappa_2 e^{2i\Phi}[\Delta_1 + i(\gamma_1 - \kappa_1)]}{[\Delta_1 + i(\gamma_1 + \kappa_1)][\Delta_2 + i(\gamma_2 + \kappa_2)] + \kappa_1\kappa_2 e^{2i\Phi}} \quad (\text{R23})$$

$$S_{21}(\omega) = \frac{(\Delta_1 + i\gamma_1)(\Delta_2 + i\gamma_2)e^{i\Phi}}{[\Delta_1 + i(\gamma_1 + \kappa_1)][\Delta_2 + i(\gamma_2 + \kappa_2)] + \kappa_1\kappa_2e^{2i\Phi}} \quad (\text{R24})$$

$$S_{12}(\omega) = S_{21}(\omega) \quad (\text{R25})$$

$$S_{22}(\omega) = -i \frac{\kappa_1 e^{2i\Phi} [\Delta_2 + i(\gamma_2 - \kappa_2)] + \kappa_2 [\Delta_1 + i(\gamma_1 + \kappa_1)]}{[\Delta_1 + i(\gamma_1 + \kappa_1)][\Delta_2 + i(\gamma_2 + \kappa_2)] + \kappa_1\kappa_2e^{2i\Phi}} \quad (\text{R26})$$

where $\Delta_1 = \omega - \omega_1$ and $\Delta_2 = \omega - \omega_2$ are defined as frequency detunings of two magnon modes.

Poles of S-parameters (Eqs. (R23)-(R26)) occurs at zero-denominator, that is,

$$[\omega - \omega_1 + i(\gamma_1 + \kappa_1)][\omega - \omega_2 + i(\gamma_2 + \kappa_2)] + \kappa_1\kappa_2e^{2i\Phi} = 0 \quad (\text{R27})$$

Note that Eq. (R27) is just same as the eigen-equation for solving the eigenvalues of effective Hamiltonian in Eqs. (1) or Eq. (R8). Also the S-parameters in Eqs. (R23)-(R26) are equivalent to Eq. (R9).

(III) The comparison of theoretical calculations and experimental results.

We directly compare the theoretical calculations with the experimental results as shown in Fig. 3 and also copied below. As implied by the analytical formula in Eqs. (R23)-(R26), the reflection spectra contains both information about ZR (zero numerator in Eqs. (R23) and (R26)) and eigenvalues (zero denominator in Eqs. (R23)-(R26)). The real part of the calculated eigenvalues (blue dotted curves) show up as high reflection peaks which agree well with the measured data (dark region in color plot). This excellent consistency supports the validity of the effective Hamiltonian in Eq. (1) of the main text.

Duplicate of Figure 3] Perfect zero-reflection observed in experiment. a-c Mappings of $|S_{11}|$ as functions of Δ_H and Δ_2 in four cases: fixing $\omega_2/2\pi = 3.6$ GHz, 4.24 GHz, 4.69 GHz, respectively. **d** High-resolution zoom-in image of (a). **g-i** Mappings of $|S_{22}|$ as functions of Δ_H and Δ_2 in four ω_2 cases. Blue dotted curves are the fitted hybridized modes, white long dashed lines are the calculated frequencies of ZR states. The PZR conditions are marked by arrows. **j** High-resolution zoom-in image of (g). **e** and **k** are the group delay τ_g mappings corresponding to (d) and (j), the PZR emerge as singularities, where the group delay abruptly switches between negative infinity and positive infinity. **f** and **l** are the PZR spectra ($|S_{11}|$ and $|S_{22}|$) at the black arrows marked fields in (d) and (j), respectively.

The abstract and text have discussions of coherent perfect absorption- the CPA refers to the interference of counterpropagating waves leading to zero transmission and reflection. Whereas the authors bring microwaves from one port only.

Reply: We thank the Reviewer for correcting our statement of coherent perfect absorption. We agree that it is indeed misleading to readers (although in the original version we intended to use the word “coherent” to indicate the interaction between two magnon modes has the coherent coupling term at the nearly full absorption region). In the revised manuscript, we have improved the statement “**coherent perfect absorption/transparency**” in original manuscript Line 22 to “**nearly full absorption/transmission (NFA/NFT)**”. In original manuscript Lines 73-75, the statement “**By tuning the ZR condition on or off resonance with the eigenstates, we can control the absorption/transmission in a coherent or decoherent manner.**” has been improved in revised manuscript, as “**This reflection singularity of PZR is attributed to the destructive interference between the direct and high-order reflected waves. It can be regulated to coincide with or deviate from the eigenmodes and accordingly the absorption/transmission of the coupled system can be managed.**” (Page 2 Lines 77-80). That indicates if tuning the PZR condition approach to the resonant eigenstate, the nearly full absorption can be realized and the interaction between two magnon modes has coherent coupling term, while tuning the PZR condition away from the resonance of the eigenstate, two magnon modes are decoupled resulting in high transmission. In Line 82 (original manuscript), “**Structure of coherent magnonic system**” has been changed to “**Structure of the indirectly coupled two-magnon system**” (Line 85 of the revised manuscript). The legend of Fig. 1 “**Coherent magnonic system**” has been changed to “**Indirectly coupled two-magnon system**” to avoid possible confusion.

I have difficulty in understanding fig 4a- say the full black horizontal line $r_{sub1}(cal)$ —The authors previously discuss that r_{sub1} can be zero only for $\sin[2\phi]$ nonzero- but these values of ϕ are shown in the theory part. This is quite misleading.

Reply: We appreciate very much this comment and sorry for this misleading point. When $\sin(2\Phi) = 0$, the imaginary part of the ZR line becomes parallel to real axis as shown in Figs. 2e and 2g, therefore PZR does not exist. So, in Fig. 4 (a) and (b) of the revised manuscript, we have removed the special points with $\sin(2\Phi) = 0$ by using open circles and we emphasize this point in the figure caption and main text to avoid possible misleading to readers. In figure legend, “**Open circles plotted in (a, b) indicate the PZR does not exist at these points**” has been added. The related description in main text has been improved as “**The colorbar shows the hybridized states at different Φ values, at the purely coherent (LR) and purely dissipative (LA) coupling regions ($\Phi = n\pi/2, n \in N$), the PZR does not exist, and hence the full transmission cannot be reached strictly but can be approached infinitely when $\Phi \rightarrow n\pi/2$** ” see Lines 272-275 on Page 8. And we have also added a colorbar in Fig. 4a, b to clearly mark the region of LA and LR corresponding to different values of Φ .

The level attraction and repulsion discussion in lines 139 to 291 is well known and does not add anything new.

Reply: We thank the Reviewer for this comment and we agree that level attraction and repulsion has been discussed in previous literatures. Therefore, in the revised manuscript, we have made this

part more concise and stress only the difference of our work (utilizing purely indirect coupling to cover continuously from level repulsion to level attraction region). In particular, we explicitly mention that traveling photons (indirect coupling) in our work play an important role in both level attraction and level repulsion cases, unlike that in conventional wisdom traveling photons (environment) are deemed to be not involved in pure level repulsion case.

Thus in summary I do not recommend publication of the paper given that 1) the conditions for perfect zero reflection have been obtained earlier, 2) the theoretical description [eq 1] is incorrect, 3) paper has several misleading parts as described above.

Reply: We thank the Reviewer again for careful reviewing and critical comments which encourage us to largely improve our manuscript. Given that (1) the previous PZR (Fano-minima) theoretical prediction applies only for a number of *identical* and *lossless* resonators and our work provides both theoretical derivation and experimental verification and applies more generally for nonidentical and lossy resonators (unification of Fabry-Perot type and Friedrich-Wintgen type interactions and covering both nearly full transmission and nearly full absorption regions); (2) theoretical description Eq. 1 has been proved independently by the Feshbach projection approach and ABCD matrix method and agrees well with experimental data; (3) the misleading parts have been corrected in the revised version, we hope that our new version can convince the Reviewer that our work has provided robust, novel results with physically transparent and self-contained interpretations and therefore can accept our work for publication.

Reviewers' Comments:

Reviewer #1:

None

Reviewer #2:

None

Reviewer #4:

Remarks to the Author:

This manuscript presents important results on waveguide magnonics, demonstrating that the non-Hermitian nature of the interaction permits flexible control of the microwave reflection, transmission, and absorption features. Experimentally observed extremely low reflection is distinguished by the vanishing reflection amplitude and the extremely narrow bandwidth. Under the ultra-low reflectivity, this work provides a clear illustration of how virtually complete transmission and absorption can be achieved in an interfering magnonic system.

In the revised manuscript, the authors have addressed the primary concerns raised by previous reviewers and the manuscript itself is also much improved.

1. The added diagrams in Fig. 1 and Fig. 2 facilitate intuitive understanding of the waveguide magnonic system developed in this work.

2. The authors have refined certain imprecise statements, such as the effective non-Hermitian Hamiltonian (Eq. 1). They have also confirmed the validity of the effective Hamiltonian and provided additional information about the derivation of Eq. 1 in the supplementary information. The erroneous notes in Fig. 4 have been amended.

3. The authors have added some background literature, such as a distinct theoretical scheme to achieve zero reflection and their previous works on coherent and dissipative coupling in cavity magnonics. By comparing the present work to the previous ones, the authors appropriately emphasize the significance and novelty of the new study. Indeed, this paper demonstrates both theoretically and experimentally that perfect zero-reflection can be realized via the indirect coupling between two non-identical lossy resonators. In addition, benefited from the flexibility in adjusting the frequency detuning and delay phase between these two magnonic resonators, the non-Hermitian interaction can be continuously tuned and utilized to control the transmission and absorption of the system.

Overall, this work provides a flexibly adjustable platform to manipulate the behavior of microwave transport and offers promising prospect of the non-Hermitian control in the areas of microwave circuits and energy harvesting. From the novelty of the work and the considerable improvement of the manuscript by addressing all questions of the three previous reviewers, I think this revised manuscript could be published in Nature Communications.

Reviewer #5:

Remarks to the Author:

The manuscript describes the characteristics of zero-reflection transmission in a non-Hermitian magnon-photon system. The authors display the perfect absorption or perfect transmission of the traveling microwave photon through the control of magnetic field and phase in a hybrid system consisting of two YIG spheres and a planar waveguide. The occurrence of the zero-reflection is accompanied by the infinite group delay. The advantageous tunability and scalability of the experimental setup have promising applications in the directors, filters, and even the quantum information process. The authors give a detailed theoretical description and demonstrate the consistence between the experimental results and the theoretical calculations. Therefore, I think this is an exciting result and worth publication in Nature Communications.

I believe that the analysis and interpretation of the data in the manuscript is strong. However, I still have a few questions about this experiment.

1. In the fabry-perot (FP-like) resonant mechanism, as shown in line of 160 of the main text, $\Phi_{\text{round}} = \Phi_1 + \Phi_2 + 2\Phi = 2n\pi$ represents the condition of a single round on the FP-like cavity. How does this change when the more circles and more resonant mode are involved?
2. In line 160 of the main text, the transmission delay phase Φ is equal to 1.07π , 1.25π , and 1.37π when $\omega_2/2\pi$ is 3.6GHz, 4.24GHz, and 4.69GHz, respectively. How is this phase calculated? Is it obtained by $\Phi = 2\pi l/\lambda$?
3. In the Fig. 3, the definition of Δ_2 has been used in the main text but it seems has not defined, although there is a definition as $\Delta_2 = \omega - \omega_2$ in supplemental information.
4. As depicted in Eq. (5), the corresponding resonant frequency of PZR is a function of the magnetic field and would also appear as a line in the spectrogram. Why does PZR appear in a certain position instead of a line in Fig. 3?
5. In line 221 of the main text, Does the figure numbering should be Fig. 2(f) and Fig. 2(h).
6. For the experimental setup, how do the YIG spheres attached to the copper waveguide, and what is the relationship between the crystallographic direction and the magnetic field. Has the dissipation of the waveguide been evaluated experimentally.

Reviewer #6:

None

Reply to Reviewer #4:

This manuscript presents important results on waveguide magnonics, demonstrating that the non-Hermitian nature of the interaction permits flexible control of the microwave reflection, transmission, and absorption features. Experimentally observed extremely low reflection is distinguished by the vanishing reflection amplitude and the extremely narrow bandwidth. Under the ultra-low reflectivity, this work provides a clear illustration of how virtually complete transmission and absorption can be achieved in an interfering magnonic system.

In the revised manuscript, the authors have addressed the primary concerns raised by previous reviewers and the manuscript itself is also much improved.

1. The added diagrams in Fig. 1 and Fig. 2 facilitate intuitive understanding of the waveguide magnonic system developed in this work.

2. The authors have refined certain imprecise statements, such as the effective non-Hermitian Hamiltonian (Eq. 1). They have also confirmed the validity of the effective Hamiltonian and provided additional information about the derivation of Eq. 1 in the supplementary information. The erroneous notes in Fig. 4 have been amended.

3. The authors have added some background literature, such as a distinct theoretical scheme to achieve zero reflection and their previous works on coherent and dissipative coupling in cavity magnonics. By comparing the present work to the previous ones, the authors appropriately emphasize the significance and novelty of the new study. Indeed, this paper demonstrates both theoretically and experimentally that perfect zero-reflection can be realized via the indirect coupling between two non-identical lossy resonators. In addition, benefited from the flexibility in adjusting the frequency detuning and delay phase between these two magnonic resonators, the non-Hermitian interaction can be continuously tuned and utilized to control the transmission and absorption of the system.

Overall, this work provides a flexibly adjustable platform to manipulate the behavior of microwave transport and offers promising prospect of the non-Hermitian control in the areas of microwave circuits and energy harvesting. From the novelty of the work and the considerable improvement of the manuscript by addressing all questions of the three previous reviewers, I think this revised manuscript could be published in Nature Communications.

Reply: We thank the Reviewer very much for the very positive comments to our revised manuscript and for supporting our work to be published.

Reply to Reviewer #5

The manuscript describes the characteristics of zero-reflection transmission in a non-Hermitian magnon-photon system. The authors display the perfect absorption or perfect transmission of the traveling microwave photon through the control of magnetic field and phase in a hybrid system consisting of two YIG spheres and a planar waveguide. The occurrence of the zero-reflection is

accompanied by the infinite group delay. The advantageous turnability and scalability of the experimental setup have promising applications in the directors, filters, and even the quantum information process. The authors give a detailed theoretical description and demonstrate the consistence between the experimental results and the theoretical calculations. Therefore, I think this is an exciting result and worth publication in Nature Communications.

I believe that the analysis and interpretation of the data in the manuscript is strong. However, I still have a few questions about this experiment.

Reply: We thank the Reviewer very much for evaluating our result to be “exciting” and supporting our work to be published. We are also very grateful to the careful additional comments which are valuable for us to optimize related description in our manuscript. We have addressed each point of the report, the detail replies are shown point-by-point as follow:

1. In the fabry-perot (FP-like) resonant mechanism, as shown in line of 160 of the main text, $\Phi_{round} = \Phi_1 + \Phi_2 + 2\Phi = 2n\pi$ represents the condition of a single round on the FP-like cavity. How does this change when the more circles and more resonant mode are involved?

Reply: We thank the Reviewer very much for this very inspiring question. Taking account of more circles and more resonant modes, the more generalized formula should be $N\Phi_{round} \equiv N(\Phi_1 + \Phi_2 + 2\Phi) = 2n\pi$ (both N and n are integers), where N denotes N times round-trip propagation of microwave photons between two resonators and n refers to the n^{th} -order of FP resonance.

This general formula can be changed into $\Phi_{round} \equiv (\Phi_1 + \Phi_2 + 2\Phi) = 2\pi(n/N)$. Let $n/N = n' + n''/N$, where n' and n'' are integers and $0 \leq n'' < N$. Then this general formula can be rewritten as $\Phi_1 + \Phi_2 + 2(\Phi - n''/N\pi) = 2n'\pi$, which means that the conclusion of our work remains valid except that the effective retardation phase is shifted to $\Phi_{eff} \equiv \Phi - n''/N\pi$.

- (i) If $n'' = 0$, then the formula reduces to our original single round-trip version and, in fact, the independence on N confirms the constructive interference of different round-trip waves, forming well-defined (n' -th order FP-like) standing wave.
- (ii) If $n'' \neq 0$, then the phase shift term ($-n''/N\pi$) may play a role in determining the coupling strength between two YIG resonators, effectively same influence as in Fig. 2 with different Φ . However, we should note that, for fixed N and n (therefore n' and n''), different phase shift terms from $0, -\pi/N, -2\pi/N, \dots, -n''\pi/N$ superimpose and contribute altogether to the wave propagation. As a consequence, typically no stable standing wave can be formed between two resonators. Experimentally, our work confirms that $n'' \neq 0$ case plays little role in our experimental data and it is sufficient to use the single round-trip formula (which includes all $n'' = 0$ contribution, i.e., $n' = n/N = 1/1, 2/2, 3/3, 4/4, \dots$ for our experiments).

We also mention that the spectral window in our practical experiments is limited (typically $\omega_0/2\pi \sim 3 - 5$ GHz), determined by ferromagnetic resonance (FMR) properties of YIG sphere. For more circles ($N \geq 2$ while $n = 1$) and more resonant modes ($n \geq 2$ while $N = 1$), the resonant frequency becomes ω_0/N and $n\omega_0$ respectively, both are out of the measurement frequency range. (Here, ω_0 is the resonant frequency of the fundamental FP mode corresponds to $\Phi_{round} \equiv (\Phi_1 + \Phi_2 + 2\Phi) = 2\pi$.)

Following Reviewer’s comment and to avoid possible confusion from readers, we have added in the line 160 of the main text that “...and this formula applies for single round-trip

propagation of microwave photons but is found to be sufficient for reproducing our following experimental data...”.

2. In line 160 of the main text, the transmission delay phase Φ is equal to 1.07π , 1.25π , and 1.37π when $\omega_2/2\pi$ is 3.6GHz, 4.24GHz, and 4.69GHz, respectively. How is this phase calculated? Is it obtained by $\Phi = 2\pi l/\lambda$?

Reply: The transmission delay phase $\Phi = 1.07\pi, 1.25\pi, 1.37\pi$ are obtained by independently fitting the dispersions in corresponding reflection mappings according to Eq. 2, for the cases $\omega/2\pi = 3.6$ GHz, 4.24 GHz, 4.69 GHz, respectively. In addition, we confirm that the obtained phase value is proportional to the corresponding frequency, i.e., $\Phi \propto \omega$. Through this linear relationship, we can derive the microwave velocity in the waveguide to be a constant $v = \omega l/\Phi = 0.17 \times 10^8$ m/s. This velocity value is as expected from the design structure of our waveguide.

Following the Reviewer’s comment, in the revised paper, line 210, we have added this explanation as “**Here, the values of Φ are obtained by fitting the dispersion of reflection mappings in Fig. 3 according to Eq. (2) (also see Supplementary Note 3, fitted curves represented by dashed lines in FIG. S3)**”.

3. In the Fig. 3, the definition of Δ_2 has been used in the main text but it seems has not defined, although there is a definition as $\Delta_2 = \omega - \omega_2$ in supplemental information.

Reply: We thank the Reviewer for this careful examination. We have added the definition of Δ_2 in main text line 254, as “**Figures 3a-c are the experimental mappings of $|S_{11}|$ as a function of $\Delta_H = \omega_1 - \omega_2$ and $\Delta_2 = \omega - \omega_2$ for the cases...**” .

4. As depicted in Eq. (5), the corresponding resonant frequency of PZR is a function of the magnetic field and would also appear as a line in the spectrogram. Why does PZR appear in a certain position instead of a line in Fig. 3?

Reply: We thank the Reviewer for this very important comment. Actually, according to $\text{Im}(\tilde{\omega}_{ZR}) = 0$, ω_1 and ω_2 in Eq. 5 are not independent but should satisfy

$$(\omega_1^2 - \omega_2^2)^2 / (\mathcal{C}'\omega_1^2 - \mathcal{C}\omega_2^2) = 4(\mathcal{C}' - \mathcal{C}), \quad (\text{R1})$$

where $\mathcal{C} = [\kappa_2(\kappa_1 - \gamma_1)\cos 2\Phi - \kappa_1(\gamma_2 + \gamma_2)]/\kappa_2\sin 2\Phi$ and $\mathcal{C}' = [-\kappa_2(\kappa_1 - \gamma_1) - \kappa_1(\gamma_2 + \kappa_2)\cos 2\Phi]/\kappa_1\sin 2\Phi$, both of which are constants in the measurement of Fig. 3. When ω_2 is fixed under a certain magnetic field, ω_1 is determined uniquely according to Eq. R1, therefore, ω_{PZR}^\pm is uniquely determined in Eq. 5 and the PZR appears at a certain position in the reflection mapping of Fig. 3. Analytically, this constraint is not easy to be incorporate into Eq. 5 in the main text due to very complex expression.

Following the reviewer’s important comment, we have added the above equation as additional constraint (Eq. S36b) in the supplementary information and improved the description of Eq. 5 in the main text line 230, as “**It’s worth noting that, in Eq. (5), ω_1 is uniquely determined by ω_2 due to the requirement of $\text{Im}(\tilde{\omega}_{ZR}) = 0$ (see Supplementary Note 2), thus the PZR appears at a certain frequency point when the external magnetic field fixed**”.

5. In line 221 of the main text, Does the figure numbering should be Fig. 2(f) and Fig. 2(h).

Reply: We thank the Reviewer for this careful examination and reminding. In line 221 of the previous manuscript, we had intended to utilize Figs. 2f and 2h to demonstrate $\text{Im}(\tilde{\omega}_{ZR})$ would

become nonzero constant and have different sign in pure LA and pure LR region ($\text{Im}(\tilde{\omega}_{ZR}) < 0$ in Fig. 2f and $\text{Im}(\tilde{\omega}_{ZR}) > 0$ in Fig. 2g), moreover, the sign change between them would induce PZR, $\text{Im}(\tilde{\omega}_{ZR}) = 0$, for instance in Fig. 2f and 2h. But we realize that the previous description may be misleading.

To avoid possible misleading, we have revised this sentence as “**In Figs. 2e and 2g, the horizontal orange lines show $\text{Im}(\tilde{\omega}_{ZR})$ become nonzero constants in pure LA (Fig. 2e) and pure LR (Fig. 2g) regions, $\text{Im}(\tilde{\omega}_{ZR}) \neq 0$ indicates there is no PZR exist. We notice that $\text{Im}(\tilde{\omega}_{ZR}) < 0$ in pure LA region and $\text{Im}(\tilde{\omega}_{ZR}) > 0$ in pure LR region, while in the general hybrid region (Figs. 2f and 2h) with $\Phi \neq n\pi/2$, there exists the critical point where the sign of $\text{Im}(\tilde{\omega}_{ZR})$ changes abruptly leading to $\text{Im}(\tilde{\omega}_{ZR}) = 0$, ensuring the general existence of PZR**”, as shown in the revised manuscript lines 223-228.

6. For the experimental setup, how do the YIG spheres attached to the copper waveguide, and what is the relationship between the crystallographic direction and the magnetic field. Has the dissipation of the waveguide been evaluated experimentally.

Reply: For the experimental setup, the YIG spheres are glued to the end of a displacement cantilever, which is connected to a three-dimensional mechanically adjustable stage. The YIG spheres are adjusted accurately close to the copper waveguide and their positions are always fixed in measurements. The crystallographic directions of the YIG spheres are not intentionally identified but remain fixed through all of our experiments. The dissipation of the waveguide can be evaluated by reflection S_{11} and transmission spectra S_{21} of bare waveguide, as shown in Figure R1. In the measured frequency range (3-5 GHz), the reflection is always below -20 dB and the insertion loss of transmission spectrum is less than 0.5 dB, as shown in the enlarged view in Figure R1.

In the revised manuscript, we have added these details in **Methods-Device description**, as “**In the measured frequency range (3-5 GHz), the insertion loss of the bare waveguide is less than 0.5 dB, and the reflection keeps below -20 dB...The crystallographic directions of two YIG spheres are not identified but remain fixed in all experiments.**” in lines 344-349.

Figure R1 | The reflection S_{11} and transmission S_{21} spectra of the bare waveguide, inset is the enlarged view of S_{21} .

Reviewers' Comments:

Reviewer #5:

Remarks to the Author:

In the revised manuscript, the authors have addressed my concerns and the manuscript itself has been much improved. I think it's suitable to be published in Nature Communications.

Reply to Reviewer #5:

Reviewer #5 (Remarks to the Author):

In the revised manuscript, the authors have addressed my concerns and the manuscript itself has been much improved. I think it's suitable to be published in Nature Communications.

Reply: We thank the Reviewer very much for the comments and for supporting our work to be published.